# User-Level Differential Privacy
# With Few Examples Per User

**Badih Ghazi**
Google Research
Mountain View, CA, US
badihghazi@gmail.com

**Pritish Kamath**
Google Research
Mountain View, CA, US
pritish@alum.mit.edu

**Ravi Kumar**
Google Research
Mountain View, CA, US
ravi.k53@gmail.com

**Pasin Manurangsi**
Google Research
Bangkok, Thailand
pasin@google.com

**Raghu Meka**
UCLA
Los Angeles, CA, US
raghum@cs.ucla.edu

**Chiyuan Zhang**
Google Research
Mountain View, CA, US
chiyuan@google.com

## Abstract

Previous work on user-level differential privacy (DP) [GKM21, BGH+23] obtained generic algorithms that work for various learning tasks. However, their focus was on the example-*rich* regime, where the users have so many examples that each user could themselves solve the problem. In this work we consider the example-*scarce* regime, where each user has only a few examples, and obtain the following results:

- For approximate-DP, we give a generic transformation of any item-level DP algorithm to a user-level DP algorithm. Roughly speaking, the latter gives a (multiplicative) savings of $O_{\varepsilon,\delta}(\sqrt{m})$ in terms of the number of users required for achieving the same utility, where $m$ is the number of examples per user. This algorithm, while recovering most known bounds for specific problems, also gives new bounds, e.g., for PAC learning.

- For pure-DP, we present a simple technique for adapting the exponential mechanism [MT07] to the user-level setting. This gives new bounds for a variety of tasks, such as private PAC learning, hypothesis selection, and distribution learning. For some of these problems, we show that our bounds are near-optimal.

## 1 Introduction

Differential privacy (DP) [DMNS06, DKM+06] has become a widely popular notion of privacy quantifying a model's leakage of personal user information. It has seen a variety of industrial and governmental deployments including [Gre16, App17, DKY17, Abo18, RE19, TM20]. Many fundamental private machine learning algorithms in literature have been devised under the (implicit) assumption that each user only contributes a single example to the training dataset, which we will refer to as *item-level DP*. In practice, each user often provides multiple examples to the training data. In this *user-level DP* setting, the output of the algorithm is required to remain (approximately) statistically indistinguishable even when all the items belonging to a single user are substituted (formal definitions below). Given such a strong (and preferable) privacy requirement, it is highly non-trivial for the algorithm to take advantage of the increased number of examples. Several recent studies have explored this question and showed—for a wide variety of tasks—that, while challenging, this is possible [LSY+20, LSA+21, GKM21, ILPS22, BGH+23].

To discuss further, let us define some notation. We use $n$ to denote the number of users and $m$ to denote the number of examples per user. We use $\mathcal{Z}$ to denote the universe of possible samples. We write $\boldsymbol{x} = (x_{1,1}, \ldots, x_{1,m}, x_{2,1}, \ldots, x_{n,m}) \in \mathcal{Z}^{nm}$ to denote the input to our algorithm, and

37th Conference on Neural Information Processing Systems (NeurIPS 2023).

$\boldsymbol{x}_i := (x_{i,1}, \ldots, x_{i,m})$ for $i \in [n]$ to denote the $i$th user's examples. Furthermore, we write $\boldsymbol{x}_{-i}$ to denote the input with the $i$th user's data $\boldsymbol{x}_i$ removed. Similarly, for $S \subseteq [n]$, we write $\boldsymbol{x}_{-S}$ to denote the input with all the examples of users in $S$ removed.

**Definition 1** (User-Level Neighbors). Two inputs $\boldsymbol{x}, \boldsymbol{x}'$ are *user-level neighbors*, denoted by $\boldsymbol{x} \asymp \boldsymbol{x}'$, iff $\boldsymbol{x}_{-i} = \boldsymbol{x}'_{-i}$ for some $i \in [n]$.

**Definition 2** (User-Level and Item-Level DP). For $\varepsilon, \delta > 0$, we say that a randomized algorithm[1] $\mathbb{A} : \mathcal{Z} \to \mathcal{O}$ is $(\varepsilon, \delta)$-*user-level DP* iff, for any $\boldsymbol{x} \asymp \boldsymbol{x}'$ and any $S \subseteq \mathcal{O}$, we have $\Pr[\mathbb{A}(\boldsymbol{x}) \in S] \leq e^\varepsilon \Pr[\mathbb{A}(\boldsymbol{x}') \in S] + \delta$. When $m = 1$, we say that $\mathbb{A}$ is $(\varepsilon, \delta)$-*item-level DP*.

The case $\delta = 0$ is referred to as *pure-DP*, whereas the case $\delta > 0$ is referred to as *approximate-DP*.

To formulate statistical tasks studied in our work, we consider the setting where there is an unknown distribution $\mathcal{D}$ over $\mathcal{Z}$ and the input $\boldsymbol{x} \sim \mathcal{D}^{nm}$ consists of $nm$ i.i.d. samples drawn from $\mathcal{D}$. A task $\mathfrak{T}$ (including the desired accuracy) is defined by $\Psi_{\mathfrak{T}}(\mathcal{D})$, which is the set of "correct" answers. For a parameter $\gamma$, we say that the algorithm $\mathbb{A}$ is $\gamma$-*useful* if $\Pr_{\boldsymbol{x} \sim \mathcal{D}^{nm}}[\mathbb{A}(\boldsymbol{x}) \in \Psi_{\mathfrak{T}(\mathcal{D})}] \geq \gamma$ for all valid $\mathcal{D}$. Furthermore, we say that $nm$ is the *sample complexity* of the algorithm, whereas $n$ is its *user complexity*. For $m \in \mathbb{N}$ and $\varepsilon, \delta > 0$, let $n_m^{\mathfrak{T}}(\varepsilon, \delta; \gamma)$ denote the smallest user complexity of any $(\varepsilon, \delta)$-user-level DP algorithm that is $\gamma$-useful for $\mathfrak{T}$. When $\gamma$ is not stated, it is assumed to be $2/3$.

**Generic Algorithms Based on Stability.** So far, there have been two main themes of research on user-level DP learning. The first aims to provide generic algorithms that work with many tasks. Ghazi et al. [GKM21] observed that any *pseudo-globally stable* (aka *reproducible* [ILPS22]) algorithm can be turned into a user-level DP algorithm. Roughly speaking, pseudo-global stability requires that the algorithm, given a random input dataset and a random string, returns a canonical output—which may depend on the random string—with a large probability (e.g., $0.5$). They then show that the current best generic PAC learning item-level DP algorithms from [GGKM21] (which are based on yet another notion of stability) can be made into pseudo-globally stable algorithms. This transforms the aforementioned item-level DP algorithms into user-level DP algorithms.

In a recent breakthrough, Bun et al. [BGH+23] significantly expanded this transformation by showing that *any* item-level DP algorithm can be compiled into a pseudo-globally stable algorithm—albeit with some overhead in the sample complexity. Combining with [GKM21], they then get a generic transformation of any item-level DP algorithm to a user-level DP algorithm. Such results are of (roughly) the following form: If there is an item-level DP algorithm for some task with sample complexity $n$, then there is a user-level DP algorithm with user complexity $O(\log(1/\delta)/\varepsilon)$ as long as $m \geq \widetilde{\Omega}_{\varepsilon,\delta}(n^2)$. In other words, if each user has sufficiently many samples, the algorithm needs very few users to learn under the user-level DP constraint. However, the requirement on the number of examples per user can be prohibitive: due to the nature of the reduction, it requires each user to have enough samples to learn by themselves. This leads to the following natural question:[2]

> *Is there a generic transformation from item-level to user-level DP algorithms for small $m$?*

**Algorithms for Specific Tasks & the $\sqrt{m}$ Savings.** Meanwhile, borrowing a page from its item-level DP counterpart, another active research theme has been to study specific tasks and provide (tight) upper and lower bounds on their sample complexity for user-level DP. Problems such as mean estimation, stochastic convex optimization (SCO), and discrete distribution learning are well-understood under user-level DP (for approximate-DP); see, e.g., [LSY+20, LSA+21, NME22, GKK+23]. A pattern has emerged through this line of studies: (the privacy-dependent part of) the user complexity in the user-level DP setting is often roughly $1/\sqrt{m}$ smaller than that of the item-level DP setting. For example, for the task of discrete distribution learning on domain $[k]$ up to total variation distance of $\alpha > 0$ (denoted by $\mathrm{DD}k(\alpha)$), the user complexity for the user-level DP is [DHS15, ASZ21, NME22]

$$n_m^{\mathrm{DD}k(\alpha)}(\varepsilon, \delta) = \tilde{\Theta}_\delta \left( \frac{k}{\varepsilon \alpha \sqrt{m}} + \frac{k}{\alpha^2 m} \right).$$

---

[1]For simplicity, we will only consider the case where the output space $\mathcal{O}$ is finite throughout this work to avoid measure-theoretic issues. This assumption is often without loss of generality since even the output in continuous problems can be discretized in such a way that the "additional error" is negligible.

[2]Of course, there is a generic transformation where each user throws away all but one example; here, we are looking for one that can reduce the user complexity compared to the item-level setting.

We can see that the first privacy-dependent term decreases by a factor of $\sqrt{m}$ whereas the latter privacy-independent term decreases by a factor of $m$. A similar phenomenon also occurs in the other problems discussed above. As such, it is intriguing to understand this question further:

*Is there a common explanation for these $\sqrt{m}$ saving factors for the privacy-dependent term?*

## 1.1 Our Contributions

**Approximate-DP.** We answer both questions by showing a generic way to transform any item-level DP algorithm into a user-level DP algorithm such that the latter saves roughly $\sqrt{m}$ in terms of the number of users required. Since the formal expression is quite involved, we state below a simplified version, which assumes that the sample complexity can be written as the sum of two terms, the privacy-independent term (i.e., $n_{\text{stat}}^{\mathfrak{T}}$) and the privacy-dependent term that grows linearly in $1/\varepsilon$ and polylogarithmically in $1/\delta$ (i.e., $\left( \frac{\log(1/\delta)^{O(1)}}{\varepsilon} \right) \cdot n_{\text{priv}}^{\mathfrak{T}}$). This is the most common form of sample complexity of DP learning discovered so far in the literature—in particular, this covers all the tasks we have discussed previously. The corollary below shows that the privacy-dependent term decreases by a factor of roughly $\sqrt{m}$ whereas the privacy-independent term decreases by a factor of $m$. The full version of the statement can be found in Theorem 10.

**Corollary 3** (Main Theorem–Simplified). *Let $\mathfrak{T}$ be any task and $\gamma > 0$ be a parameter. Suppose that for any sufficiently small $\varepsilon, \delta > 0$, there is an $(\varepsilon, \delta)$-item-level DP algorithm that is $\gamma$-useful for $\mathfrak{T}$ with sample complexity $\widetilde{O}\left( \frac{\log(1/\delta)^c}{\varepsilon} \cdot n_{\text{priv}}^{\mathfrak{T}} + n_{\text{stat}}^{\mathfrak{T}} \right)$ for some constant c. Then, for any sufficiently small $\varepsilon, \delta > 0$, there exists an $(\varepsilon, \delta)$-user-level DP algorithm that is $(\gamma - o(1))$-useful for $\mathfrak{T}$ and has user complexity*

$$\widetilde{O}\left( \frac{1}{\sqrt{m}} \cdot \frac{\log(1/\delta)^{c+3/2}}{\varepsilon^2} \cdot n_{\text{priv}}^{\mathfrak{T}} + \frac{1}{m} \cdot n_{\text{stat}}^{\mathfrak{T}} + \frac{\log(1/\delta)}{\varepsilon} \right).$$

Our result provides a powerful and generic tool in learning with user-level DP, as it can translate any item-level DP bound to user-level DP bound. Specifically, up to polylogarithmic terms and the dependency on $\varepsilon$, the corollary above recovers all the aforementioned user complexity bounds for mean estimation, SCO, and discrete distribution learning. In addition, it also presents new sample complexity bounds for the moderate $m$ regime for PAC learning.

While our result is very general, it—perhaps inevitably—comes at a cost: the algorithm is computationally inefficient and the dependence on $\varepsilon$ (and $\delta$) is not tight. We discuss this—along with other open questions—in more detail in Section 5.

**Pure-DP.** Surprisingly, the pure-DP setting is less explored compared to approximate-DP in the context of learning with user-level DP; the sample complexity of several of the problems mentioned above (e.g., discrete distribution learning and SCO) is well-understood under approximate-DP but not so under pure-DP. While we do not provide a generic transformation from item-level DP algorithms to user-level DP algorithms for pure-DP, we give simple modifications of the popular pure-DP *exponential mechanism* [MT07] that allow it to be used in the user-level DP setting. Consequently, we obtain improved bounds for several problems. Due to space constraints, in the main body of the paper, we will only discuss PAC learning. Results for other problems, such as hypothesis testing and distribution learning can be found in Appendix E.

In PAC learning, $\mathcal{Z} = \mathcal{X} \times \{0, 1\}$ and there is a concept class $\mathcal{C} \subseteq \{0, 1\}^{\mathcal{X}}$. An error of $c : \mathcal{Z} \to \{0, 1\}$ w.r.t. $\mathcal{D}$ is defined as $\text{err}_{\mathcal{D}}(c) := \Pr_{(x,y) \sim \mathcal{D}}[c(x) \neq y]$. The task $\text{PAC}(\mathcal{C}; \alpha)$ is to, for any distribution $\mathcal{D}$ that has zero error on some concept $c^* \in \mathcal{C}$ (aka *realizable by* $\mathcal{C}$), output $c : \mathcal{Z} \to \{0, 1\}$ such that $\text{err}_{\mathcal{D}}(c) \leq \alpha$. We give a tight characterization of the user complexity for PAC learning[3]:

**Theorem 4.** *Let $\mathcal{C}$ be any concept class with probabilistic representation dimension $d$ (i.e. $\text{PRDim}(\mathcal{C}) = d$). Then, for any sufficiently small $\alpha, \varepsilon > 0$ and for all $m \in \mathbb{N}$, we have*

$$n_m^{\text{PAC}(\mathcal{C}; \alpha)}(\varepsilon, \delta = 0) = \tilde{\Theta}\left( \frac{d}{\varepsilon} + \frac{d}{\varepsilon \alpha m} \right).$$

Previous work [GKM21, BGH$^+$23] achieves the tight sample complexity only for large $m$ ($\gg d^2$).

---

[3]PRDim is defined in Definition 20.

#### 1.1.1 Technical Overview

In this section, we give a rough overview of our proofs. We will be intentionally vague; all definitions and results will be formalized later in the paper.

**Approximate-DP.** At a high-level, our proof proceeds roughly as follows. First, we show that any $(\varepsilon, \delta)$-*item-level* DP $\mathbb{A}$ with high probability satisfies a local version of user-level DP for which we only compare[4] $\mathbb{A}(\boldsymbol{x}_{-i})$ for different $i$, but with $\varepsilon$ that is increased by a factor of roughly $\sqrt{m}$ (see Definition 16 and Theorem 17). If this were to hold not only for $\boldsymbol{x}_{-i}$ but also for all user-level neighbors $\boldsymbol{x}'$ of $\boldsymbol{x}$, then we would have been done since the $(\varepsilon, \delta)$-item-level DP algorithm would have also yielded $(\varepsilon\sqrt{m}, \delta)$-*user-level* DP (which would then imply a result in favor of Corollary 3). However, this is not true in general and is where the second ingredient of our algorithm comes in: We use a propose-test-release style algorithm [DL09] to check whether the input is close to a "bad" input for which the aforementioned local condition does not hold. If so, then we output $\bot$; otherwise, we run the item-level DP algorithm. Note that the test passes w.h.p. and thus we get the desired utility.

This second step is similar to recent work of Kohli and Laskowski [KL21] and Ghazi et al. [GKK+23], who used a similar algorithm for additive noise mechanisms (namely Laplace and Gaussian mechanisms). The main difference is that instead of testing for local *sensitivity* of the function as in [KL21, GKK+23], we directly test for the privacy loss of the algorithm $\mathbb{A}$.

As for the first step, we prove this via a connection to a notion of generalization called *sample perfect generalization* [CLN+16] (see Definition 12). Roughly speaking, this notion measures the privacy loss when we run an $(\varepsilon, \delta)$-item-level DP algorithm on two $\boldsymbol{x}, \boldsymbol{x}'$ drawn independently from $\mathcal{D}^n$. We show that w.h.p. $\mathbb{A}(\boldsymbol{x}) \approx_{\varepsilon', \delta'} \mathbb{A}(\boldsymbol{x}')$ for $\varepsilon' = O(\sqrt{n \log(1/\delta)} \cdot \varepsilon)$. Although ostensibly unrelated to the local version of user-level DP that we discussed above, it turns out that these are indeed related: if we view the algorithm that fixes examples of all but one user, then applying the sample perfect generalization bound on just the input of the last user (which consists of only $m$ samples) shows that the privacy loss when changing the last user is $O(\sqrt{m \log(1/\delta)} \cdot \varepsilon)$ w.h.p. It turns out that we can turn this argument into a formal bound for our local notion of user-level DP (see Section 3.2.1.)

We remark that our results on sample perfect generalization also resolve an open question of Cummings et al. [CLN+16]; we defer this discussion to Appendix C.1.

Finally, we also note that, while Bun et al. [BGH+23] also uses the sample perfect generalization notion, their conversion from item-level DP algorithms to user-level DP algorithms require constructing pseudo-globally stable algorithms with sample complexity $m$. This is impossible when $m$ is small, which is the reason why their reduction works only in the example-rich setting.

**Pure-DP.** Recall that the exponential mechanism [MT07] works as follows. Suppose there is a set $\mathcal{H}$ of candidates we would like to select from, together with the scoring functions $(\mathrm{scr}_H)_{H \in \mathcal{H}}$, where $\mathrm{scr}_H : \mathcal{Z}^n \to \mathbb{R}$ has sensitivity at most $\Delta$. The algorithm then outputs $H$ with probability proportional to $\exp\left(-\varepsilon \cdot \mathrm{scr}_H(\boldsymbol{x})/2\Delta\right)$. The error guarantee of the algorithm scales with the sensitivity $\Delta$ (see Theorem 9.) It is often the case that $\mathrm{scr}_H$ depends on the sum of individual terms involving each item. Our approach is to clip the contribution from each user so that the sensitivity is small. We show that selecting the clipping threshold appropriately is sufficient obtain new user-level pure-DP (and in some cases optimal) algorithms.

## 2 Preliminaries

Let $[n] = \{1, \ldots, n\}$, let $[x]_+ = \max(0, x)$, and let $\widetilde{O}(f)$ denote $O(f \log^c f)$ for some constant $c$. Let $\mathrm{supp}(A)$ denote the support of distribution $A$. We recall some standard tools from DP [Vad17].

For two distributions $A, B$, let $d_\varepsilon(A \parallel B) = \sum_{x \in \mathrm{supp}(A)}[A(x) - e^\varepsilon B(x)]_+$ denote the $e^\varepsilon$-*hockey stick divergence*. For brevity, we write $A \approx_{\varepsilon, \delta} B$ to denote $d_\varepsilon(A \parallel B) \le \delta$ and $d_\varepsilon(B \parallel A) \le \delta$.

**Lemma 5** ("Triangle Inequality"). *If $A \approx_{\varepsilon', \delta'} B$ and $B \approx_{\varepsilon, \delta} C$, then $A \approx_{\varepsilon + \varepsilon', e^\varepsilon \delta' + e^{\varepsilon'} \delta} C$.*

**Lemma 6** (Group Privacy). *Let $\mathbb{A}$ be any $(\varepsilon, \delta)$-item-level DP algorithm and $\boldsymbol{x}, \boldsymbol{x}'$ be $k$-neighbors[5], then $\mathbb{A}(\boldsymbol{x}) \approx_{\varepsilon', \delta'} \mathbb{A}(\boldsymbol{x}')$ where $\varepsilon' = k\varepsilon, \delta' = \frac{e^{k\varepsilon}-1}{e^\varepsilon-1}\delta \le ke^{k\varepsilon}\delta$.*

---

[4]In Theorem 17, we need to compare $\mathbb{A}(\boldsymbol{x}_{-S})$ for $S$'s of small sizes as well.

[5]I.e., there exists $\boldsymbol{x} = \boldsymbol{x}_0, \boldsymbol{x}_1, \ldots, \boldsymbol{x}_k = \boldsymbol{x}'$ such that $\boldsymbol{x}_{i-1}, \boldsymbol{x}_i$ are item-level neighbors for all $i \in [k]$.

**Theorem 7** (Amplification-by-Subsampling [BBG18]). *Let $n, n' \in \mathbb{N}$ be such that $n' \geq n$. For any $(\varepsilon, \delta)$-item-level DP algorithm $\mathbb{A}$ with sample complexity $n$, let $\mathbb{A}'$ denote the algorithm with sample complexity $n'$ that randomly chooses $n$ out of the $n'$ input samples and then runs $\mathbb{A}$ on the subsample. Then $\mathbb{A}'$ is $(\varepsilon', \delta')$-DP where $\varepsilon' = \ln\left(1 + \eta(e^\varepsilon - 1)\right), \delta' = \eta\delta$ for $\eta = n/n'$.*

We also need the definition of the shifted and truncated discrete Laplace distribution:

**Definition 8** (Shifted Truncated Discrete Laplace Distribution). *For any $\varepsilon, \delta > 0$, let $\kappa = \kappa(\varepsilon, \delta) := 1 + \lceil \ln(1/\delta)/\varepsilon \rceil$ and let $\mathsf{TDLap}(\varepsilon, \delta)$ be the distribution supported on $\{0, \ldots, 2\kappa\}$ with probability mass function at $x$ being proportional to $\exp\left(-\varepsilon \cdot |x - \kappa|\right)$.*

**Exponential Mechanism (EM).** In the (generalized) *selection* problem (SELECT), we are given a candidate set $\mathcal{H}$ and scoring functions $\mathrm{scr}_H$ for all $H \in \mathcal{H}$ whose sensitivities are at most $\Delta$. We say that an algorithm $\mathbb{A}$ is $(\alpha, \beta)$-accurate iff $\Pr_{H \sim \mathbb{A}(\boldsymbol{x})}[\mathrm{scr}_H(\boldsymbol{x}) \leq \min_{H' \in \mathcal{H}} \mathrm{scr}_{H'}(\boldsymbol{x}) + \alpha] \geq 1 - \beta$.

**Theorem 9** ([MT07]). *There is an $\varepsilon$-DP $(O(\Delta \cdot \log(|\mathcal{H}|/\beta)/\varepsilon), \beta)$-accurate algorithm for SELECT.*

# 3 Approximate-DP

In this section, we prove our main result on approximate-DP user-level learning, which is stated in Theorem 10 below. Throughout the paper we say that a parameter is "sufficiently small" if it is at most an implicit absolute constant.

**Theorem 10** (Main Theorem). *Let $\mathfrak{T}$ be any task and $\gamma > 0$ be a parameter. Then, for any sufficiently small $\varepsilon, \delta > 0$ and $m \in \mathbb{N}$, there exists $\varepsilon' = \frac{\varepsilon^2}{\log(1/\delta)\sqrt{m\log(m/\delta)}}$ and $\delta' = \Theta\left(\delta \cdot \frac{\varepsilon}{m\log(1/\delta)}\right)$ such that*

$$n_m^{\mathfrak{T}}(\varepsilon, \delta; \gamma - o(1)) \leq \widetilde{O}\left(\frac{\log(1/\delta)}{\varepsilon} + \frac{1}{m} \cdot n_1^{\mathfrak{T}}(\varepsilon', \delta'; \gamma)\right).$$

Note that Corollary 3 follows directly from Theorem 10 by plugging in the expression $n_1^{\mathfrak{T}}(\varepsilon', \delta'; \gamma) = \widetilde{O}\left(\frac{\log(1/\delta')^c}{\varepsilon'} \cdot n_{\mathrm{priv}}^{\mathfrak{T}} + n_{\mathrm{stat}}^{\mathfrak{T}}\right)$ into the bound.

While it might be somewhat challenging to interpret the bound in Theorem 10, given that the $\varepsilon$ values on the left and the right hand sides are not the same, a simple subsampling argument (similar to one in [BGH$^+$23]) allows us to make the LHS and RHS have the same $\varepsilon$. In this case, we have that the user complexity is reduced by a factor of roughly $1/\sqrt{m}$, as stated below. We remark that the $\delta$ values are still different on the two sides; however, if we assume that the dependence on $1/\delta$ is only polylogarithmic, this results in at most a small gap of at most polylogarithmic factor in $1/\delta$.

**Theorem 11** (Main Theorem–Same $\varepsilon$ Version). *Let $\mathfrak{T}$ be any task and $\gamma > 0$ be a parameter. Then, for any sufficiently small $\varepsilon, \delta > 0$, and $m \in \mathbb{N}$, there exists $\delta' = \Theta\left(\delta \cdot \frac{\varepsilon}{m\log(1/\delta)}\right)$ such that*

$$n_m^{\mathfrak{T}}(\varepsilon, \delta; \gamma - o(1)) \leq \widetilde{O}\left(\frac{\log(1/\delta)}{\varepsilon} + \frac{1}{\sqrt{m}} \cdot \frac{\log(1/\delta)^{1.5}}{\varepsilon} \cdot n_1^{\mathfrak{T}}(\varepsilon, \delta'; \gamma)\right).$$

Even though Theorem 11 might be easier to interpret, we note that it does not result in as sharp a bound as Theorem 10 in some scenarios. For example, if we use Theorem 11 under the assumption in Corollary 3, then the privacy-independent part would be $\frac{1}{\sqrt{m}} \cdot \frac{\log(1/\delta)^{1.5}}{\varepsilon} \cdot n_{\mathrm{stat}}^{\mathfrak{T}}$, instead of $\frac{1}{m} \cdot n_{\mathrm{stat}}^{\mathfrak{T}}$ implied by Theorem 10. (On the other hand, the privacy-dependent part is the same.)

The remainder of this section is devoted to the proof of Theorem 10. The high-level structure follows the overview in Section 1.1.1: first we show in Section 3.1 that any item-level DP algorithm satisfies sample perfect generalization. Section 3.2.1 then relates this to a local version of user-level DP. We then describe the full algorithm and its guarantees in Section 3.2.2. Since we fix a task $\mathfrak{T}$ throughout this section, we will henceforth discard the superscript $\mathfrak{T}$ for brevity.

## 3.1 DP Implies Sample Perfect Generalization

We begin with the definition of *sample perfect generalization* algorithms.

**Definition 12** ([CLN+16]). *For $\beta, \varepsilon, \delta > 0$, an algorithm $\mathbb{A} : \mathcal{Z}^n \to \mathcal{O}$ is said to be $(\beta, \varepsilon, \delta)$-sample perfectly generalizing iff, for any distribution $\mathcal{D}$ (over $\mathcal{Z}$), $\Pr_{\boldsymbol{x}, \boldsymbol{x}' \sim \mathcal{D}^n}[\mathbb{A}(\boldsymbol{x}) \approx_{\varepsilon, \delta} \mathbb{A}(\boldsymbol{x}')] \geq 1 - \beta$.*

The main result of this subsection is that any $(\varepsilon, \delta)$-item-level DP algorithm is $(\beta, O(\varepsilon\sqrt{n \log(1/\beta\delta)}), O(n\delta))$-sample perfectly generalizing:

**Theorem 13** (DP $\Rightarrow$ Sample Perfect Generalization). *Suppose that $\mathbb{A} : \mathcal{Z}^n \to \mathcal{O}$ is $(\varepsilon, \delta)$-item-level DP, and assume that $\varepsilon, \delta, \beta, \varepsilon\sqrt{n \log(1/\beta\delta)} > 0$ are sufficiently small. Then, $\mathbb{A}$ is $(\beta, \varepsilon', \delta')$-sample perfectly generalizing, where $\varepsilon' \leq O(\varepsilon\sqrt{n \log(1/\beta\delta)})$ and $\delta' = O(n\delta)$.*

### 3.1.1 Bounding the Expected Divergence

As a first step, we upper bound the expectation of the hockey stick divergence $d_{\varepsilon'}(\mathbb{A}(\boldsymbol{x}) \parallel \mathbb{A}(\boldsymbol{x}'))$:

**Lemma 14.** *Suppose that $\mathbb{A} : \mathcal{Z}^n \to \mathcal{O}$ is an $(\varepsilon, \delta)$-item-level DP algorithm. Further, assume that $\varepsilon, \delta$, and $\varepsilon\sqrt{n \log(1/\delta)} > 0$ are sufficiently small. Then, for $\varepsilon' = O(\varepsilon\sqrt{n \log(1/\delta)})$, we have*

$$\mathbb{E}_{\boldsymbol{x}, \boldsymbol{x}' \sim \mathcal{D}^n}[d_{\varepsilon'}(\mathbb{A}(\boldsymbol{x}) \parallel \mathbb{A}(\boldsymbol{x}'))] \leq O(n\delta).$$

This proof follows an idea from [CLN+16], who prove a similar statement for *pure-DP* algorithms (i.e., $\delta = 0$). For every output $o \in \mathcal{O}$, they consider the function $g^o : \mathcal{Z}^n \to \mathbb{R}$ defined by $g^o(\boldsymbol{x}) := \ln(\Pr[\mathbb{A}(\boldsymbol{x}) = o])$. The observation here is that, in the pure-DP case, this function is $\varepsilon$-Lipschitz (i.e., changing a single coordinate changes its value by at most $\varepsilon$). They then apply McDiarmid's inequality on $g^o$ to show that the function values are well-concentrated with a variance of $O(\sqrt{n} \cdot \varepsilon)$. This suffices to prove the above statement for the case where the algorithm is pure-DP (but the final bound still contains $\delta$). To adapt this proof to the approximate-DP case, we prove a "robust" version of McDiarmid's inequality that works even for the case where the Lipschitz property is violated on some pairs of neighbors (Lemma 32). This inequality is shown by arguing that we may change the function "slightly" to make it multiplicative-Lipschitz, which then allows us to apply standard techniques. Due to space constraints, we defer the full proof, which is technically involved, to Appendix C.3.

### 3.1.2 Boosting the Probability: Proof of Theorem 13

While Lemma 14 bounds the expectation of the hockey stick divergence, it is insufficient to get arbitrarily high probability bound (i.e., for any $\beta$ as in Theorem 13); for example, using Markov's inequality would only be able to handle $\beta > \delta'$. However, we require very small $\beta$ in a subsequent step of the proofs (in particular Theorem 17). Fortunately, we observe that it is easy to "boost" the $\beta$ to be arbitrarily small, at an additive cost of $\varepsilon\sqrt{n \log(1/\beta)}$ in the privacy loss.

To do so, we will need the following concentration result of Talagrand[6] for product measures:

**Theorem 15** ([Tal95]). *For any distribution $\mathcal{D}$ over $\mathcal{Z}$, $\mathcal{E} \subseteq \mathcal{Z}^n$, and $t > 0$, if $\Pr_{\boldsymbol{x} \sim \mathcal{D}^n}[\boldsymbol{x} \in \mathcal{E}] \geq 1/2$, then $\Pr_{\boldsymbol{x} \sim \mathcal{D}^n}[\boldsymbol{x} \in \mathcal{E}_{\leq t}] \geq 1 - 0.5 \exp\left(-t^2/4n\right)$, where $\mathcal{E}_{\leq t} := \{\boldsymbol{x}' \in \mathcal{Z}^n \mid \exists \boldsymbol{x} \in \mathcal{E}, \|\boldsymbol{x}' - \boldsymbol{x}\|_0 \leq t\}$.*

*Proof of Theorem 13.* We consider two cases based on whether $\beta \geq 2^{-n}$ or not. Let us start with the case that $\beta \geq 2^{-n}$. From Lemma 14, for $\varepsilon'' = O(\varepsilon\sqrt{n \log(1/\delta)})$, we have

$$\mathbb{E}_{\boldsymbol{x}, \boldsymbol{x}' \sim \mathcal{D}^n}[d_{\varepsilon''}(\mathbb{A}(\boldsymbol{x}) \parallel \mathbb{A}(\boldsymbol{x}'))] \leq O(n\delta) =: \delta''.$$

Let $\mathcal{E} \subseteq \mathcal{Z}^n \times \mathcal{Z}^n$ denote the set of $(\tilde{\boldsymbol{x}}, \tilde{\boldsymbol{x}}')$ such that $\mathbb{A}(\tilde{\boldsymbol{x}}) \approx_{\varepsilon'', 4\delta''} \mathbb{A}(\tilde{\boldsymbol{x}}')$. By Markov's inequality, we have that $\Pr_{\boldsymbol{x}, \boldsymbol{x}' \sim \mathcal{D}^n}[d_{\varepsilon''}(\mathbb{A}(\tilde{\boldsymbol{x}}) \parallel \mathbb{A}(\tilde{\boldsymbol{x}}')) \leq 4\delta''] \geq 3/4$ and $\Pr_{\boldsymbol{x}, \boldsymbol{x}' \sim \mathcal{D}^n}[d_{\varepsilon''}(\mathbb{A}(\tilde{\boldsymbol{x}}') \parallel \mathbb{A}(\tilde{\boldsymbol{x}})) \leq 4\delta''] \geq 3/4$ and hence $\Pr_{\boldsymbol{x}, \boldsymbol{x}' \sim \mathcal{D}^n}[(\boldsymbol{x}, \boldsymbol{x}') \in \mathcal{E}] \geq 1/2$. By Theorem 15, for $t = O(\sqrt{n \log(1/\beta)})$, we have

$$\Pr_{\boldsymbol{x}, \boldsymbol{x}' \sim \mathcal{D}^n}[(\boldsymbol{x}, \boldsymbol{x}') \in \mathcal{E}_{\leq t}] \geq 1 - \beta.$$

Next, consider any $(\boldsymbol{x}, \boldsymbol{x}') \in \mathcal{E}_{\leq t}$. Since there exists $(\tilde{\boldsymbol{x}}, \tilde{\boldsymbol{x}}') \in \mathcal{E}$ such that $\|\boldsymbol{x} - \tilde{\boldsymbol{x}}\|_0, \|\boldsymbol{x}' - \tilde{\boldsymbol{x}}'\|_0 \leq t$, we may apply Lemma 6 to conclude that $d_{\varepsilon t}(\mathbb{A}(\boldsymbol{x}) \parallel \mathbb{A}(\tilde{\boldsymbol{x}})), d_{\varepsilon t}(\mathbb{A}(\tilde{\boldsymbol{x}}') \parallel \mathbb{A}(\boldsymbol{x}')) \leq$

---

[6]This is a substantially simplified version compared to the one in [Tal95], but is sufficient for our proof.

$O(t\delta)$. Furthermore, since $d_{\varepsilon''}(\mathbb{A}(\tilde{\boldsymbol{x}}) \parallel \mathbb{A}(\tilde{\boldsymbol{x}}')) \le 4\delta''$, we may apply Lemma 5 to conclude that $d_{\varepsilon'}(\mathbb{A}(\boldsymbol{x}) \parallel \mathbb{A}(\boldsymbol{x}')) \le \delta'$ where $\varepsilon' = \varepsilon'' + 2t\varepsilon = O(\varepsilon\sqrt{n\log(1/\beta\delta)})$ and $\delta' = O(\delta'' + t\delta) = O(n\delta)$. Similarly, we also have $d_{\varepsilon'}(\mathbb{A}(\boldsymbol{x}' \parallel \mathbb{A}(\boldsymbol{x})) \le \delta'$. In other words, $\mathbb{A}(\boldsymbol{x}) \approx_{\varepsilon',\delta'} \mathbb{A}(\boldsymbol{x}')$. Combining this and the above inequality implies that $\mathbb{A}$ is $(\beta, \varepsilon', \delta')$-sample perfectly generalizing as desired.

For the case $\beta < 2^{-n}$, we may immediately apply group privacy (Lemma 6). Since any $\boldsymbol{x}, \boldsymbol{x}' \sim \mathcal{D}^n$ satisfies $\|\boldsymbol{x} - \boldsymbol{x}'\|_0 \le n$, we have $\mathbb{A}(\boldsymbol{x}) \approx_{\varepsilon',\delta'} \mathbb{A}(\boldsymbol{x}')$ for $\varepsilon' = n\varepsilon$ and $\delta' = O(n\delta)$. This implies that $\mathbb{A}$ is $(\beta, \varepsilon', \delta')$-sample perfectly generalizing (in fact, $(0, \varepsilon', \delta')$-sample perfectly generalizing), which concludes our proof. $\qquad\square$

## 3.2 From Sample Perfect Generalization to User-Level DP

Next, we will use the sample perfect generalization result from the previous subsection to show that our algorithm satisfies a certain definition of a local version of user-level DP. We then finally turn this intuition into an algorithm and prove Theorem 10.

### 3.2.1 Achieving Local-Deletion DP

We start by defining *local-deletion DP* (for user-level DP). As alluded to earlier, this definition only considers $\boldsymbol{x}_{-i}$ for different $i$'s. We also define the multi-deletion version where we consider $\boldsymbol{x}_{-S}$ for all subsets $S$ of small size.

**Definition 16** (Local-Deletion DP). *An algorithm $\mathbb{A}$ is $(\varepsilon, \delta)$-local-deletion DP (abbreviated LDDP) at input $\boldsymbol{x}$ if $\mathbb{A}(\boldsymbol{x}_{-i}) \approx_{\varepsilon,\delta} \mathbb{A}(\boldsymbol{x}_{-i'})$ for all $i, i' \in [n]$. Furthermore, for $r \in \mathbb{N}$, an algorithm $\mathbb{A}$ is $(r, \varepsilon, \delta)$-local-deletion DP at $\boldsymbol{x}$ if $\mathbb{A}(\boldsymbol{x}_{-S}) \approx_{\varepsilon,\delta} \mathbb{A}(\boldsymbol{x}_{-S'})$ for all $S, S' \subseteq [n]$ such that $|S| = |S'| = r$.*

Below we show that, with high probability (over the input $\boldsymbol{x}$), any item-level DP algorithm satisfies LDDP with the privacy loss increased by roughly $r\sqrt{m}$. The proof uses the crucial observation that relates sample perfect generalization to user-level DP.

**Theorem 17.** *Let $n, r \in \mathbb{N}$ with $r \le n$, and let $\mathbb{A} : \mathcal{X}^{(n-r)m} \to \mathcal{O}$ be any $(\varepsilon', \delta')$-item-level DP algorithm. Further, assume that $\varepsilon', \delta', \varepsilon' r \sqrt{m \log(n/\beta\delta')} > 0$ are sufficiently small. Then, for any distribution $\mathcal{D}$,*

$$\Pr_{\boldsymbol{x} \sim \mathcal{D}^{nm}}[\mathbb{A} \text{ is } (r, \varepsilon, \delta)\text{-LDDP at } \boldsymbol{x}] \ge 1 - \beta,$$

*where $\varepsilon = O(\varepsilon' r \sqrt{m \log(n/\beta\delta')})$ and $\delta = O(rm\delta')$.*

*Proof.* Let $\beta' = \beta/\binom{n}{r}^2$. Consider any fixed $S, S' \subseteq [n]$ such that $|S| = |S'| = r$. Let $T = S \cup S', t = |T|, U = S' \setminus S, U' = S \setminus S', q = |U| = |U'|$, and let $\mathbb{A}'_{\boldsymbol{x}_{-T}}$ denote the algorithm defined by $\mathbb{A}'_{\boldsymbol{x}_{-T}}(\boldsymbol{y}) = \mathbb{A}(\boldsymbol{x}_{-T} \cup \boldsymbol{y})$.

Thus, for any fixed $\boldsymbol{x}_{-T} \in \mathcal{Z}^{(n-t)m}$, Theorem 13 implies that

$$\Pr_{\boldsymbol{x}_U, \boldsymbol{x}_{U'} \sim \mathcal{D}^{qm}}\left[\mathbb{A}'_{\boldsymbol{x}_{-T}}(\boldsymbol{x}_U) \not\approx_{\varepsilon,\delta} \mathbb{A}'_{\boldsymbol{x}_{-T}}(\boldsymbol{x}_{U'})\right] \le \beta', \tag{1}$$

where $\varepsilon = O(\varepsilon'\sqrt{qm\log(1/\beta'\delta')}) \le O(\varepsilon' r\sqrt{m\log(n/\beta\delta')})$ and $\delta = O(rm\delta')$.

Notice that $\mathbb{A}'_{\boldsymbol{x}_{-T}}(\boldsymbol{x}_U) = \mathbb{A}(\boldsymbol{x}_{-S})$ and $\mathbb{A}'_{\boldsymbol{x}_{-T}}(\boldsymbol{x}_{U'}) = \mathbb{A}(\boldsymbol{x}_{-S'})$. Thus, we have

$$\Pr_{\boldsymbol{x} \sim \mathcal{D}^{nm}}[\mathbb{A}(\boldsymbol{x}_{-S}) \not\approx_{\varepsilon,\delta} \mathbb{A}(\boldsymbol{x}_{-S'})] = \mathbb{E}_{\boldsymbol{x}_{-T} \sim \mathcal{D}^{(n-t)m}}\left[\Pr_{\boldsymbol{x}_U, \boldsymbol{x}_{U'} \sim \mathcal{D}^{qm}}\left[\mathbb{A}'_{\boldsymbol{x}_{-T}}(\boldsymbol{x}_U) \not\approx_{\varepsilon,\delta} \mathbb{A}'_{\boldsymbol{x}_{-T}}(\boldsymbol{x}_{U'})\right]\right]$$

$$\overset{(1)}{\le} \mathbb{E}_{\boldsymbol{x}_{-T} \sim \mathcal{D}^{(n-t)m}}[\beta'] = \beta'.$$

Hence, by taking a union bound over all possible $S, S' \subseteq [n]$ such that $|S| = |S'| = r$, we have $\Pr_{\boldsymbol{x} \sim \mathcal{D}^{nm}}[\mathbb{A} \text{ is not } (r, \varepsilon, \delta)\text{-LDDP at } \boldsymbol{x}] \le \beta$ as desired. $\qquad\square$

### 3.2.2 From LDDP to DP via Propose-Test-Release

Our user-level DP algorithm is present in Algorithm 1. Roughly speaking, we try to find a small subset $S$ such that $\boldsymbol{x}_{-S}$ is LDDP with appropriate parameters. (Throughout, we assume that $n \ge 4\kappa$

where $\kappa$ is from Definition 8.) The target size of $S$ is noised using the discrete Laplace distribution. As stated in Section 1.1.1, the algorithm is very similar to that of [GKK$^+$23], with the main difference being that (i) $\mathcal{X}_{\text{stable}}^{R_1}$ is defined based on LDDP instead of the local deletion sensitivity and (ii) our output is $\mathbb{A}(\boldsymbol{x}_{-T})$, whereas their output is the function value with Gaussian noise added. It is not hard to adapt their proof to show that our algorithm is $(\varepsilon, \delta)$-user-level DP, as stated below. (Full proof is deferred to Appendix C.4.)

**Lemma 18** (Privacy Guarantee). DelStab$_{\varepsilon,\delta,\mathbb{A}}$ *is* $(\varepsilon, \delta)$-*user-level DP.*

---

**Algorithm 1** DelStab$_{\varepsilon,\delta,\mathbb{A}}(\boldsymbol{x})$

---

1: **Input:** Dataset $\boldsymbol{x} \in \mathcal{X}^{nm}$
2: **Parameters:** Privacy parameters $\varepsilon, \delta$, Algorithm $\mathbb{A} : \mathcal{X}^{(n-4\kappa)m} \to \mathcal{O}$
3: $\overline{\varepsilon} \leftarrow \frac{\varepsilon}{3}, \overline{\delta} \leftarrow \frac{\delta}{e^{2\overline{\varepsilon}}+e^{\varepsilon}+2}, \kappa \leftarrow \kappa(\overline{\varepsilon}, \overline{\delta})$.
4: Sample $R_1 \sim \mathsf{TDLap}(\overline{\varepsilon}, \overline{\delta})$                            {See Definition 8}
5: $\mathcal{X}_{\text{stable}}^{R_1} \leftarrow \left\{ S \subseteq [n] : |S| = R_1, \mathbb{A} \text{ is } (4\kappa - R_1, \overline{\varepsilon}, \overline{\delta})\text{-LDDP at } \boldsymbol{x}_{-S} \right\}$     {See Definition 16}
6: **if** $|\mathcal{X}_{\text{stable}}^{R_1}| = \emptyset$ **then**
7:     **return** $\perp$
8: Choose $S \in \mathcal{X}_{\text{stable}}^{R_1}$ uniformly at random
9: Choose $T \supseteq S$ of size $4\kappa$ uniformly at random
10: **return** $\mathbb{A}(\boldsymbol{x}_{-T})$

---

Next, we prove the utility guarantee under the assumption that $\mathbb{A}$ is item-level DP:

**Lemma 19** (Utility Guarantee). *Let* $\varepsilon, \delta > 0$ *be sufficiently small. Suppose that* $\mathbb{A}$ *is* $(\varepsilon', \delta')$-*item-level DP such that* $\varepsilon' = \Theta\left( \frac{\varepsilon}{\kappa\sqrt{m \log(nm/\delta)}} \right)$ *and* $\delta' = \Theta\left( \frac{\delta}{\kappa m} \right)$. *If* $\mathbb{A}$ *is* $\gamma$-*useful for any task* $\mathfrak{T}$ *with* $(n - 4\kappa)m$ *samples, then* DelStab$_{\varepsilon,\delta,\mathbb{A}}$ *is* $(\gamma - o(1))$-*useful for* $\mathfrak{T}$ *(with* $nm$ *samples).*

*Proof.* Consider another algorithm $\mathbb{A}_0$ defined in the same way as DelStab$_{\varepsilon,\delta,\mathbb{A}}$ except that on Line 5, we simply let $\mathcal{X}_{\text{stable}}^{R_1} \leftarrow \binom{[n]}{R_1}$. Note that $\mathbb{A}_0(\boldsymbol{x})$ is equivalent to randomly selecting $n - 4\kappa$ users and running $\mathbb{A}$ on it. Therefore, we have $\Pr_{\boldsymbol{x} \sim \mathcal{D}^{nm}}[\mathbb{A}_0(\boldsymbol{x}) \in \Psi_{\mathfrak{T}}] = \Pr_{\boldsymbol{x} \sim \mathcal{D}^{(n-4\kappa)m}}[\mathbb{A}(\boldsymbol{x}) \in \Psi_{\mathfrak{T}}] \geq \gamma$

Meanwhile, $\mathbb{A}_0$ and DelStab$_{\varepsilon,\delta,\mathbb{A}}$ are exactly the same as long as $\mathcal{X}_{\text{stable}}^{R_1}(\boldsymbol{x}) = \binom{[n]}{R_1}$ in the latter. In other words, we have

$$\Pr_{\boldsymbol{x} \sim \mathcal{D}^{nm}}[\mathsf{DelStab}_{\varepsilon,\delta,\mathbb{A}}(\boldsymbol{x}) \in \Psi_{\mathfrak{T}}] \geq \Pr_{\boldsymbol{x} \sim \mathcal{D}^{nm}}[\mathbb{A}_0(\boldsymbol{x}) \in \Psi_{\mathfrak{T}}] - \Pr_{\boldsymbol{x} \sim \mathcal{D}^{nm}}\left[ \mathcal{X}_{\text{stable}}^{R_1}(\boldsymbol{x}) \neq \binom{[n]}{R_1} \right]$$
$$\geq \gamma - \Pr_{\boldsymbol{x} \sim \mathcal{D}^{nm}}[\mathbb{A} \text{ is not } (4\kappa, \overline{\varepsilon}, \overline{\delta})\text{-LDDP at } \boldsymbol{x}] \qquad \geq \gamma - o(1),$$

where in the last inequality we apply Theorem 17 with, e.g., $\beta = 0.1/nm$.                            $\square$

**Putting Things Together.** Theorem 10 is now "almost" an immediate consequence of Lemmas 18 and 19 (using $\kappa = O\left(\log(1/\delta)/\varepsilon\right)$). The only challenge here is that $\varepsilon'$ required in Lemma 19 depends polylogarithmically on $n$. Nonetheless, we show below via a simple subsampling argument that this only adds polylogarithmic overhead to the user/sample complexity.

*Proof of Theorem 10.* Let $\varepsilon' = \varepsilon'(n), \delta'$ be as in Lemma 19. Furthermore, let $\varepsilon'' = \frac{\varepsilon^2}{\log(1/\delta)\sqrt{m \log(m/\delta)}}$. Notice that $\varepsilon'(n)/\varepsilon'' \leq (\log n)^{O(1)}$. Let $N \in \mathbb{N}$ be the smallest number such that $N$ is divisible by $m$ and

$$\left(e^{\varepsilon''} - 1\right) \cdot \frac{n_1(\varepsilon'', \delta')}{N} \leq e^{\varepsilon'(N/m + 4\kappa)} - 1,$$

Observe that $N \leq \widetilde{O}\left(m\kappa + n_1(\varepsilon'', \delta')\right)$.

From the definition of $n_1$, there exists an $(\varepsilon'', \delta')$-item-level DP algorithm $\mathbb{A}_0$ that is $\gamma$-useful for $\mathfrak{T}$ with sample complexity $n_1(\varepsilon'', \delta')$. We start by constructing an algorithm $\mathbb{A}$ on $N$ samples as follows: randomly sample (without replacement) $n_1(\varepsilon'', \delta')$ out of $N$ items, then run $\mathbb{A}_0$ on this

subsample. By amplification-by-subsampling (Theorem 7), we have that $\mathbb{A}$ is $(\varepsilon'(n), \delta')$-item-level DP for $n = N/m + 4\kappa$. Thus, by Lemma 18 and Lemma 19, we have that $\mathsf{DelStab}_{\varepsilon,\delta,\mathbb{A}}$ is $(\varepsilon, \delta)$-DP and $(\gamma - o(1))$-useful for $\mathfrak{T}$. Its user complexity is

$$n = N/m + 4\kappa = \widetilde{O}\left(\frac{\log(1/\delta)}{\varepsilon} + \frac{1}{m} \cdot n_1\left(\frac{\varepsilon^2}{\log(1/\delta)\sqrt{m\log(m/\delta)}}, \delta'\right)\right). \qquad \square$$

## 4 Pure-DP: PAC Learning

In this section we prove Theorem 4. Let $\boldsymbol{z}$ denote the input and for $i \in [n]$, let $\boldsymbol{z}_i = ((x_{i,1}, y_{i,1}), \ldots, (x_{i,m}, y_{i,m}))$ denote the input to the $i$th user. A *concept class* is a set of functions of the form $\mathcal{X} \to \{0, 1\}$. For $T \subseteq \mathcal{X} \times \{0, 1\}$, we say that it is *realizable* by $c : \mathcal{X} \to \{0, 1\}$ iff $c(x) = y$ for all $(x, y) \in T$. Recall the definition of PRDim:

**Definition 20** ([BNS19])**.** A distribution $\mathcal{P}$ on concept classes is an $(\alpha, \beta)$-*probabilistic representation* (abbreviated as $(\alpha, \beta)$-*PR*) of a concept class $\mathcal{C}$ if for every $c \in \mathcal{C}$ and for every distribution $\mathcal{D}_{\mathcal{X}}$ on $\mathcal{X}$, with probability $1 - \beta$ over $\mathcal{H} \sim \mathcal{P}$, there exists $h \in \mathcal{H}$ such that $\Pr_{x \sim \mathcal{D}_{\mathcal{X}}}[c(x) \neq h(x)] \leq \alpha$.

The $(\alpha, \beta)$-*PR dimension* of $\mathcal{C}$ is defined as $\mathrm{PRDim}_{\alpha,\beta}(\mathcal{C}) := \min_{(\alpha,\beta)\text{-PR } \mathcal{P} \text{ of } \mathcal{C}} \mathrm{size}(\mathcal{P})$, where $\mathrm{size}(\mathcal{P}) := \max_{\mathcal{H} \in \mathrm{supp}(\mathcal{P})} \log |\mathcal{H}|$. Furthermore, let the *probabilistic representation dimension* of $\mathcal{C}$ be $\mathrm{PRDim}(\mathcal{C}) := \mathrm{PRDim}_{1/4,1/4}(\mathcal{C})$.

**Lemma 21** ([BNS19])**.** *For any $\mathcal{C}$ and $\alpha \in (0, 1/2)$, $\mathrm{PRDim}_{\alpha,1/4}(\mathcal{C}) \leq \widetilde{O}(\mathrm{PRDim}(\mathcal{C}) \cdot \log(1/\alpha))$.*

*Proof of Theorem 4.* The lower bound $\tilde{\Omega}(d/\varepsilon)$ was shown in [GKM21] (for any value of $m$). The lower bound $\tilde{\Omega}\left(\frac{d}{\varepsilon\alpha m}\right)$ follows from the lower bound of $\tilde{\Omega}\left(\frac{d}{\varepsilon\alpha}\right)$ on the sample complexity in the item-level setting by [BNS19] and the observation that any $(\varepsilon, \delta)$-user-level DP algorithm is also $(\varepsilon, \delta)$-item-level DP with the same sample complexity. Thus, we may focus on the upper bound.

It suffices to only prove the upper bound $\widetilde{O}\left(\frac{d}{\varepsilon} + \frac{d}{\varepsilon\alpha m}\right)$ assuming[7] $m \leq 1/\alpha$. Let $\mathcal{P}$ denote a $(0.01\alpha, 1/4)$-PR of $\mathcal{C}$; by Lemma 21, there exists such $\mathcal{P}$ with $\mathrm{size}(\mathcal{P}) \leq \widetilde{O}(d \cdot \log(1/\alpha))$. Assume that the number $n$ of users is at least $\frac{\kappa \cdot \mathrm{size}(\mathcal{P})}{\varepsilon\alpha m} = \widetilde{O}\left(\frac{d}{\varepsilon\alpha m}\right)$, where $\kappa$ is a sufficiently large constant. Our algorithm works as follows: Sample $\mathcal{H} \sim \mathcal{P}$ and then run the $\varepsilon$-DP EM (Theorem 9) on candidate set $\mathcal{H}$ with the scoring function $\mathrm{scr}_h(\boldsymbol{z}) := \sum_{i \in [n]} \mathbf{1}[\boldsymbol{z}_i \text{ is not realizable by } h]$.

Observe that the sensitivity of $\mathrm{scr}_h$ is one. Indeed, this is where our "clipping" has been applied: a standard item-level step would sum up the error across all the samples, resulting in the sensitivity as large as $m$, while our scoring function above "clips" the contribution of each user to just one.

A simple concentration argument (deferred to Appendix D) gives us the following:

**Lemma 22.** *With probability 0.9 (over the input), the following hold for all $h \in \mathcal{H}$:*

*(i) If $\mathrm{err}_{\mathcal{D}}(h) \leq 0.01\alpha$, then $\mathrm{scr}_h(\boldsymbol{z}) \leq 0.05\alpha nm$.*
*(ii) If $\mathrm{err}_{\mathcal{D}}(h) > \alpha$, then $\mathrm{scr}_h(\boldsymbol{z}) > 0.1\alpha nm$.*

We now assume that the event in Lemma 22 holds and finish the proof. Since a $\mathcal{P}$ is a $(0.01\alpha, 1/4)$-PR of $\mathcal{C}$, with probability $3/4$, we have that there exists $h^* \in \mathcal{H}$ such that $\mathrm{err}_{\mathcal{D}}(h^*) \leq 0.01\alpha$. From Lemma 22(i), we have $\mathrm{scr}_{h^*}(\boldsymbol{z}) \leq 0.05\alpha nm$. Thus, by Theorem 9, with probability $2/3$, the algorithm outputs $h$ that satisfies $\mathrm{scr}_h(\boldsymbol{z}) \leq 0.05\alpha nm + O(\mathrm{size}(\mathcal{P})/\varepsilon)$, which, for a sufficiently large $\kappa$, is at most $0.1\alpha nm$. By Lemma 22(ii), this implies that $\mathrm{err}_{\mathcal{D}}(h) \leq \alpha$ as desired. $\qquad \square$

## 5 Conclusion and Open Questions

We presented generic techniques to transform item-level DP algorithms to user-level DP algorithms.

---

[7]If $m$ is larger, then we can simply disregard the extra examples (as the first term dominates the bound).

There are several open questions. Although our upper bounds are demonstrated to be tight for many tasks discussed in this work, it is not hard to see that they are not always tight[8]; thus, an important research direction is to get a better understanding of the tight user complexity bounds for different learning tasks.

On a more technical front, our transformation for approximate-DP (Theorem 10) has an item-level privacy loss (roughly) of the form $\frac{\varepsilon^2}{\sqrt{m \log(1/\delta)^3}}$ while it seems plausible that $\frac{\varepsilon}{\sqrt{m \log(1/\delta)}}$ can be achieved; an obvious open question is to give such an improvement, or show that it is impossible. We note that the extra factor of $\log(1/\delta)/\varepsilon$ comes from the fact that Algorithm 1 uses LDDP with distance $O(\kappa) = O\left(\log(1/\delta)/\varepsilon\right)$. It is unclear how one could avoid this, given that the propose-test-release framework requires checking input at distance $O\left(\log(1/\delta)/\varepsilon\right)$.

Another limitation of our algorithms is their computational inefficiency, as their running time grows linearly with the size of the possible outputs. In fact, the approximate-DP algorithm could even be slower than this since we need to check whether $\mathbb{A}$ is LDDP at a certain input $x$ (which involves computing $d_\varepsilon(\mathbb{A}_{-S} \parallel \mathbb{A}_{-S'})$). Nevertheless, given the generality of our results, it is unlikely that a generic efficient algorithm exists with matching user complexity; for example, even the algorithm for user-level DP-SCO in [GKK+23] has a worst case running time that is not polynomial. Thus, it remains an interesting direction to devise more efficient algorithms for specific tasks.

Finally, while our work has focused on the central model of DP, where the algorithm gets the access to the raw input data, other models of DP have also been studied in the context of user-level DP, such as the local DP model [GKM21, ALS23]. Here again, [GKM21, BGH+23] give a generic transformation that sometimes drastically reduces the user complexity in the example-rich case. Another interesting research direction is to devise a refined transformation for the example-sparse case (like one presented in our paper for the central model) but for the local model.

---

[8]E.g., for PAC learning, Theorem 4 provides a better user complexity than applying the approximate-DP transformation (Theorem 10) to item-level approximate-DP learners for a large regime of parameters and concept classes; the former has user complexity that decreases with $1/m$ whereas the latter decreases with $1/\sqrt{m}$.

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

## A Summary of Results

Our results for pure-DP are summarized in Table 1.

| Problem | Item-Level DP (Previous Work) | User-Level DP | |
|---|---|---|---|
| | | (Previous Work) | (Our Work) |
| PAC Learning | $\tilde{\Theta}\left(\frac{d}{\varepsilon\alpha}\right)$ [BNS19] | $\tilde{\Theta}\left(\frac{d}{\varepsilon}\right)$ when $m \geq \tilde{\Theta}\left(\frac{d^2}{\varepsilon^2\alpha^2}\right)$ [GKM21, BGH$^+$23] | $\tilde{\Theta}\left(\frac{d}{\varepsilon} + \frac{d}{\varepsilon\alpha m}\right)$ Theorem 4 |
| Agnostic PAC Learning | $\tilde{\Theta}\left(\frac{d}{\alpha^2} + \frac{d}{\varepsilon\alpha}\right)$ [BNS19] | - | $\widetilde{O}\left(\frac{d}{\varepsilon} + \frac{d}{\varepsilon\alpha\sqrt{m}} + \frac{d}{\alpha^2 m}\right)$ Theorem 40 |
| Learning Discrete Distribution | $\tilde{\Theta}\left(\frac{k}{\varepsilon\alpha} + \frac{k}{\alpha^2}\right)$ [BKSW19, ASZ21] | - | $\tilde{\Theta}\left(\frac{k}{\varepsilon} + \frac{k}{\varepsilon\alpha\sqrt{m}} + \frac{k}{\alpha^2 m}\right)$ Theorem 43 |
| Learning Product Distribution | $\tilde{\Theta}\left(\frac{kd}{\varepsilon\alpha} + \frac{kd}{\alpha^2}\right)$ [BKSW19, ASZ21] | - | $\tilde{\Theta}\left(\frac{kd}{\varepsilon} + \frac{kd}{\varepsilon\alpha\sqrt{m}} + \frac{kd}{\alpha^2 m}\right)$ Theorem 44 |
| Learning Gaussian (Known Covariance) | $\tilde{\Theta}\left(\frac{d}{\varepsilon\alpha} + \frac{d}{\alpha^2}\right)$ [BKSW19, ASZ21] | - | $\tilde{\Theta}\left(\frac{d}{\varepsilon} + \frac{d}{\varepsilon\alpha\sqrt{m}} + \frac{d}{\alpha^2 m}\right)$ Theorem 45 |
| Learning Gaussian (Bounded Cov.) | $\tilde{\Theta}\left(\frac{d^2}{\varepsilon\alpha} + \frac{d^2}{\alpha^2}\right)$ [BKSW19, KLSU19] | - | $\tilde{\Theta}\left(\frac{d^2}{\varepsilon} + \frac{d^2}{\varepsilon\alpha\sqrt{m}} + \frac{d^2}{\alpha^2 m}\right)$ Theorem 46 |

Table 1: **Summary of results on pure-DP.** For (agnostic) PAC learning, $d$ denotes the probabilistic representation dimension (Definition 20) of the concept class. To the best of our knowledge, none of the distribution learning problems has been studied explicitly under pure user-level DP in previous work, although some bounds can be derived using previous techniques for approximate-DP.

For approximate-DP, our transformation (Theorem 10) is very general and can be applied to any statistical tasks. Due to this, we do not attempt to exhaustively list its corollaries. Rather, we give only a few examples of consequences of Theorem 10 in Table 2. We note that in all cases, we either improve upon previous results or match them up to $\frac{\log(1/\delta)^{O(1)}}{\varepsilon}$ factor for all values of $m$.

| Problem | Item-Level DP (Previous Work) | User-Level DP | |
|---|---|---|---|
| | | (Previous Work) | (Our Work) |
| PAC Learning | $\widetilde{O}\left(\frac{d_L^6}{\varepsilon\alpha^2}\right)$ [GGKM21] | $\widetilde{O}\left(\frac{1}{\varepsilon}\right)$ when $m \geq \tilde{\Theta}\left(\frac{d_L^{12}}{\varepsilon^2\alpha^4}\right)$ [GGKM21, BGH$^+$23] | $\widetilde{O}\left(\frac{1}{\varepsilon} + \frac{d_L^6}{\varepsilon^2\alpha^2\sqrt{m}}\right)$ Theorem 10 |
| Learning Discrete Distribution | $\tilde{\Theta}\left(\frac{k}{\varepsilon\alpha} + \frac{k}{\alpha^2}\right)$ [BKSW19, ASZ21] | $\tilde{\Theta}\left(\frac{1}{\varepsilon} + \frac{k}{\varepsilon\alpha\sqrt{m}} + \frac{k}{\alpha^2 m}\right)$ [NME22] | $\widetilde{O}\left(\frac{1}{\varepsilon} + \frac{k}{\varepsilon^2\alpha\sqrt{m}} + \frac{k}{\alpha^2 m}\right)$ Theorem 10 |
| Learning Product Distribution | $\widetilde{O}\left(\frac{kd}{\varepsilon\alpha} + \frac{kd}{\alpha^2}\right)$ [BKSW19] | - | $\widetilde{O}\left(\frac{1}{\varepsilon} + \frac{kd}{\varepsilon^2\alpha\sqrt{m}} + \frac{kd}{\alpha^2 m}\right)$ Theorem 10 |

Table 2: **Example results on approximate-DP.** Each of our results is a combination of Theorem 10 and the previously known item-level DP bound. For PAC learning, $d_L$ denotes the Littlestone's dimension of the concept class. We assume that $\delta \ll \varepsilon$ and use $\widetilde{O}, \tilde{\Theta}$ to also hide polylogarithmic factors in $1/\delta$.

## B Additional Preliminaries

In this section, we list some additional material that is required for the remaining proofs.

For $a, b \in \mathbb{R}$ such that $a \leq b$, let $\mathrm{clip}_{a,b} : \mathbb{R} \to \mathbb{R}$ denote the function

$$\mathrm{clip}_{a,b}(x) = \begin{cases} a & \text{if } x < a, \\ b & \text{if } x > b, \\ x & \text{otherwise.} \end{cases}$$

For two distributions $A, B$, we use $d_{\mathrm{tv}}(A, B) := \frac{1}{2} \sum_{x \in \mathrm{supp}(A) \cup \mathrm{supp}(B)} |A(x) - B(x)|$ to denote their total variation distance, $d_{\mathrm{KL}}(A \parallel B) := \sum_{x \in \mathrm{supp}(A)} A(x) \cdot \log(A(x)/B(x))$ to denote their Kullback–Leibler divergence, and $d_{\chi^2}(A \parallel B) := \sum_{x \in \mathrm{supp}(A)} \frac{(A(x) - B(x))^2}{B(x)}$ to denote the $\chi^2$-divergence. We use $A^k$ to denote the distribution of $k$ independent samples drawn from $A$. The following three well-known facts will be useful in our proofs.

**Lemma 23.** *For any two distributions $A$ and $B$,*

*(i) (Pinsker's inequality) $d_{\mathrm{tv}}(A, B) \leq \sqrt{\frac{1}{2} \cdot d_{\mathrm{KL}}(A \parallel B)}$.*

*(ii) $d_{\mathrm{KL}}(A^k \parallel B^k) = k \cdot d_{\mathrm{KL}}(A \parallel B)$ for $k \in \mathbb{N}$.*

*(iii) $d_{\mathrm{KL}}(A \parallel B) \leq d_{\chi^2}(A \parallel B)$.*

We use $\|u\|_0$ to denote the *Hamming "norm"*[9] of $u$, i.e., the number of non-zero coordinates of $u$. The following is a well-known fact (aka *constant-rate constant-distance error correcting codes*), which will be useful in our lower bound proofs.

**Theorem 24.** *For $d \in \mathbb{N}$, there are $u_1, \ldots, u_W \in \{0, 1\}^d$ for $W = 2^{\Omega(d)}$ such that $\|u_i - u_j\|_0 \geq \Omega(d)$ for all distinct $i, j \in [W]$.*

### B.1 Concentration Inequalities

Next, we list a few concentration inequalities in convenient forms for subsequent proofs.

**Chernoff bound.** We start with the standard Chernoff bound.

**Lemma 25** (Chernoff Bound). *Let $X_1, \ldots, X_n$ denote i.i.d. Bernoulli random variables with success probability $p$. Then, for all $\delta > 0$, we have*

$$\Pr[X_1 + \cdots + X_n \leq (1 - \delta)pn] \leq \exp\left(\frac{-\delta^2 pn}{2}\right),$$

*and,*

$$\Pr[X_1 + \cdots + X_n \geq (1 + \delta)pn] \leq \exp\left(\frac{-\delta^2 pn}{2 + \delta}\right).$$

**McDiarmid's inequality.** For a distribution $\mathcal{D}$ on $\mathcal{Z}$, and a function $g : \mathcal{Z}^n \to \mathbb{R}$, let $\mu_{\mathcal{D}^n}(g) := \mathbb{E}_{\boldsymbol{x} \sim \mathcal{D}^n}[g(\boldsymbol{x})]$. We say that $g$ is a *$c$-bounded difference* function iff $|g(\boldsymbol{x}) - g(\boldsymbol{x}')| \leq c$ for all $\boldsymbol{x}, \boldsymbol{x}' \in \mathcal{Z}^n$ that differ on a single coordinate (i.e., $\|\boldsymbol{x} - \boldsymbol{x}'\|_0 = 1$).

**Lemma 26** (McDiarmid's inequality). *Let $g : \mathcal{Z}^n \to \mathbb{R}$ be any $c$-bounded difference function. Then, for all $t > 0$*

$$\Pr[|g(\boldsymbol{x}) - \mu_{\mathcal{D}^n}(g)| > t] \leq 2 \exp\left(\frac{-2t^2}{nc^2}\right).$$

We will also use the following consequence of Azuma's inequality.

**Lemma 27.** *Let $C_1, \ldots, C_n$ be sequence of random variables. Suppose that $|C_i| \leq \alpha$ almost surely and that $\gamma \leq \mathbb{E}[C_i \mid C_1 = c_1, \ldots, C_{i-1} = c_{i-1}] \leq 0$ for any $c_1, \ldots, c_{i-1}$. Then, for every $G > 0$, we have $\Pr\left[n\gamma - \alpha\sqrt{nG} \leq \sum_{i \in [n]} C_i \leq \alpha\sqrt{nG}\right] \geq 1 - 2\exp(-G/2)$.*

---

[9]This is not actually a norm, but often called as such.

### B.2 Observations on Item-level vs User-level DP

Here we list a couple of trivial observations on the user / sample complexity in the item vs user-level DP settings.

**Observation 28.** *Let $\mathfrak{T}$ be any task and $\gamma > 0$ be a parameter. Then, for any $\varepsilon, \delta \geq 0$, and $m \in \mathbb{N}$, it holds that*

*1. $n_m^{\mathfrak{T}}(\varepsilon, \delta; \gamma) \leq n_1^{\mathfrak{T}}(\varepsilon, \delta; \gamma)$, and*
*2. $n_m^{\mathfrak{T}}(\varepsilon, \delta; \gamma) \geq \lceil n_1^{\mathfrak{T}}(\varepsilon, \delta; \gamma)/m \rceil$.*

*Proof.* Both parts follow by simple reductions as given below.

1. Let $\mathbb{A}$ be an $(\varepsilon, \delta)$-item-level DP $\gamma$-useful algorithm for $\mathfrak{T}$. Then, in the user-level DP algorithm, each user can simply throw away all-but-one example and run $\mathbb{A}$. Clearly, this algorithm is $(\varepsilon, \delta)$-user-level DP and is $\gamma$-useful for $\mathfrak{T}$. The user complexity remains the same.
2. Let $\mathbb{A}$ be an $(\varepsilon, \delta)$-user-level DP $\gamma$-useful algorithm for $\mathfrak{T}$ with user complexity $n_m^{\mathfrak{T}}(\varepsilon, \delta; \gamma)$. Then, in the item-level DP algorithm, if there are $N = m \cdot n_m^{\mathfrak{T}}(\varepsilon, \delta; \gamma)$ users, we can just run $\mathbb{A}$ by grouping each $m$ examples together into a "super-user". Clearly, this algorithm is also $(\varepsilon, \delta)$-item-level DP and has user complexity $m \cdot n_m^{\mathfrak{T}}(\varepsilon, \delta; \gamma)$. Thus, we have $m \cdot n_m^{\mathfrak{T}}(\varepsilon, \delta; \gamma) \leq n_1^{\mathfrak{T}}(\varepsilon, \delta; \gamma)$. □

## C   Missing Details from Section 3

### C.1   DP Implies Perfect Generalization

We start by recalling the definition of *perfectly generalizing* algorithms.

**Definition 29** (Perfect generalization [CLN⁺16, BF16])**.** For $\beta, \varepsilon, \delta > 0$, an algorithm $\mathbb{A} : \mathcal{Z}^n \to \mathcal{O}$ is said to be $(\beta, \varepsilon, \delta)$-*perfectly generalizing* iff, for any distribution $\mathcal{D}$ (over $\mathcal{Z}$), there exists a distribution $\mathrm{SIM}_{\mathcal{D}}$ such that

$$\Pr_{\boldsymbol{x} \sim \mathcal{D}^n}[\mathbb{A}(\boldsymbol{x}) \approx_{\varepsilon, \delta} \mathrm{SIM}_{\mathcal{D}}] \geq 1 - \beta.$$

It is known that perfect generalization and sample perfect generalization (Definition 12) are equivalent:

**Lemma 30** ([CLN⁺16, BGH⁺23])**.** *Any $(\beta, \varepsilon, \delta)$-perfectly generalizing algorithm is also $(\beta, 2\varepsilon, 3\delta)$-sample perfectly generalizing. Any $(\beta, \varepsilon, \delta)$-sample perfectly generalizing algorithm is also $(\sqrt{\beta}, \varepsilon, \delta + \sqrt{\beta})$-perfectly generalizing; furthermore, this holds even when we set $\mathrm{SIM}_{\mathcal{D}}$ to be the distribution of $\mathbb{A}(\boldsymbol{x})$ where $\boldsymbol{x} \sim \mathcal{D}^m$.*

Thus, by plugging the above into Theorem 13, we immediately arrive at the following theorem, which shows that any $(\varepsilon, \delta)$-DP algorithm is $(\beta, \widetilde{O}(\varepsilon\sqrt{n}), O(n\delta))$-perfectly generalizing. This resolves an open question of [CLN⁺16].

**Theorem 31** (DP implies perfect generalization)**.** *Suppose that $\mathbb{A}$ is an $(\varepsilon, \delta)$-item-level DP algorithm. Further, assume that $\varepsilon, \delta, \beta$, and $\varepsilon\sqrt{n \log(1/\beta\delta)}$ are sufficiently small. Then, $\mathbb{A}$ is $(\beta, \varepsilon', \delta')$-perfectly generalizing, where $\varepsilon' \leq O(\varepsilon\sqrt{n \log(1/\beta\delta)})$ and $\delta' = O(n\delta)$.*

*Furthermore, this holds even when we set $\mathrm{SIM}_{\mathcal{D}}$ to be the distribution of $\mathbb{A}(\boldsymbol{x})$ where $\boldsymbol{x} \sim \mathcal{D}^n$.*

We remark that Bun et al. [BGH⁺23] showed that any $(\varepsilon, \delta)$-DP algorithm can be "repackaged" to an $(\beta, \varepsilon', \delta')$-perfectly generalizing algorithm with similar parameters as our theorem above. However, unlike our result, their proof does *not* show that the original algorithm is perfectly generalizing.

### C.2   Achieving the Same $\varepsilon$ via Subsampling: Proof of Theorem 11

*Proof of Theorem 11.* Let $\varepsilon'' = \frac{\varepsilon^2}{\log(1/\delta)\sqrt{m \log(m/\delta)}}$. We claim that $n_1(\varepsilon'', \delta') \leq n_1(\varepsilon, \delta') \cdot O(\varepsilon/\varepsilon'')$; the lemma follows from this claim by simply plugging this bound into Theorem 10.

To see that this claim holds, by definition of $n_1$, there exists an $(\varepsilon, \delta')$-item-level DP algorithm $\mathbb{A}_0$ that is $\gamma$-useful for $\mathfrak{T}$ with sample complexity $n_1(\varepsilon, \delta')$. Let $N \in \mathbb{N}$ be a smallest number such that

$$(e^\varepsilon - 1)\frac{n_1(\varepsilon, \delta')}{N} \leq e^{\varepsilon''} - 1.$$

Notice that we have $N \leq n_1(\varepsilon, \delta') \cdot O(\varepsilon/\varepsilon'')$. Let $\mathbb{A}$ be an algorithm on $N$ samples defined as follows: randomly sample (without replacement) $n_1(\varepsilon, \delta')$ out of $N$ samples, then run $\mathbb{A}_0$ on this subsample. By the amplification-by-subsampling theorem (Theorem 7), we have that $\mathbb{A}$ is $(\varepsilon'', \delta')$-item-level DP. It is also obvious that the output distribution of this algorithm is the same as that of $\mathbb{A}_0$ on $n_1(\varepsilon, \delta')$ examples, and this algorithm is thus also $\gamma$-useful for $\mathfrak{T}$. Thus, we have $n_1(\varepsilon'', \delta') \leq N \leq n_1(\varepsilon, \delta') \cdot O(\varepsilon/\varepsilon'')$ as desired. $\qquad\square$

## C.3 Proof of Lemma 14

### C.3.1 "Robust" McDiarmid Inequality for Almost-Multiplicative-Lipschitz function

We derive the following concentration inequality, which may be viewed as a strengthened McDiarmid's inequality (Lemma 26) for the multiplicative-Lipschitz case. To state this, let us define an additional notation: for $\boldsymbol{x} \in \mathcal{Z}^n, \tilde{x}_i \in \mathcal{Z}$, we write $\tilde{\boldsymbol{x}}^{(i)}$ to denote $\boldsymbol{x}$ but with $x_i$ replaced by $\tilde{x}_i$.

**Lemma 32.** *Let $f : \mathcal{Z}^n \to \mathbb{R}_{\geq 0}$ be any function and $\mathcal{D}$ any distribution on $\mathcal{Z}$. Assume that $\varepsilon, \delta > 0$, and $\varepsilon\sqrt{n\log(1/\delta)} > 0$ are sufficiently small. Then, for $\varepsilon' = O(\varepsilon\sqrt{n\log(1/\delta)})$, we have*

$$\mathbb{E}_{\boldsymbol{x},\boldsymbol{x}'\sim\mathcal{D}^n}[[f(\boldsymbol{x}) - e^{\varepsilon'} \cdot f(\boldsymbol{x}')]_+] \leq O(\mu_{\mathcal{D}^n}(f) \cdot \delta) + O\left(\sum_{i=1}^n \mathbb{E}_{\boldsymbol{x},\tilde{\boldsymbol{x}}\sim\mathcal{D}^n}\left[f(\boldsymbol{x}) - e^{\varepsilon} \cdot f(\tilde{\boldsymbol{x}}^{(i)})\right]_+\right).$$

When the second term in the RHS (i.e., $O\left(\sum_{i=1}^n \mathbb{E}_{\boldsymbol{x},\tilde{\boldsymbol{x}}\sim\mathcal{D}^n}\left[f(\boldsymbol{x}) - e^{\varepsilon} \cdot f(\tilde{\boldsymbol{x}}^{(i)})\right]_+\right)$) is zero, our lemma is (essentially) the same as McDiarmid's inequality for multiplicative-Lipschitz functions, as proved, e.g., in [CLN+16]. However, our bound is more robust, in the sense that it can handle the case where the multiplicative-Lipschitzness condition fails sometimes; this is crucial for the proof of Lemma 14 as we only assume approximate-DP.

We note that, while there are (additive) McDiarmid's inequalities that does not require Lipschitzness everywhere [Kut02, War16], we are not aware of any version that works for us due to the regime of parameters we are in and the assumptions we have. This is why we prove Lemma 32 from first principles. The remainder of this subsection is devoted to the proof of Lemma 32.

**Scalar Random Variable Adjustment for Bounded Ratio.** For the rest of this section, we write $z \approx_\varepsilon z'$ where $z, z' \in \mathbb{R}_{\geq 0}$ to indicate that $z \in [e^{-\varepsilon}z', e^\varepsilon z']$. We extend the notion similarly to $\mathbb{R}_{\geq 0}$-valued random variables: for a $\mathbb{R}_{\geq 0}$-valued random variable $Z$ and $z \in \mathbb{R}_{\geq 0}$, we write $Z \approx_\varepsilon z'$ if $z \approx_\varepsilon z'$ for each $z \in \operatorname{supp}(Z)$. Furthermore, for (possibly dependent) $\mathbb{R}_{\geq 0}$-valued random variables $Z, Z'$, we write $Z \approx_\varepsilon Z'$ to denote $z \approx_\varepsilon z'$ for all $(z, z') \in \operatorname{supp}((Z, Z'))$.

Suppose $Z$ is a $\mathbb{R}_{\geq 0}$-valued random variable such that any two values $z, \tilde{z} \in \operatorname{supp}(Z)$ satisfies $z \approx_\varepsilon \tilde{z}$. Then, we also immediately have that their mean $\mu$ is in this range and thus $Z \approx_\varepsilon \mu$. We start by showing a "robust" version of this statement, which asserts that, even if the former condition fails sometimes, we can still "move" the random variable a little bit so that the second condition holds (albeit with weaker $2\varepsilon$ bound). The exact statement is presented below.

**Lemma 33.** *Let $\varepsilon, \mu^* \geq 0$. Let $Z$ be any $\mathbb{R}_{\geq 0}$-valued random variable and $\mu := \mathbb{E}[Z]$. Then, there exists a random variable $Z'$, which is a post-processing of $Z$, such that*

*(i) $Z' \approx_{2\varepsilon} \mu^*$.*
*(ii) $\mathbb{E}[Z'] = \mu^*$.*
*(iii) $\mathbb{E}_{(Z,Z')}[|Z - Z'|] \leq |\mu - \mu^*| + 2(1 + e^{-\varepsilon}) \cdot \mathbb{E}_{Z,\tilde{Z}}\left[\left[Z - e^\varepsilon \cdot \tilde{Z}\right]_+\right]$, where $(Z, Z')$ is the canonical coupling between $Z, Z'$ (from post-processing) and $\tilde{Z}$ is an i.i.d. copy of $Z$.*

*Proof.* Let $\ell = e^{-\varepsilon} \cdot \mu, r = e^\varepsilon \cdot \mu$. We start by defining $\hat{Z} := \operatorname{clip}_{\ell,r}(Z)$. This also gives a canonical coupling between $\hat{Z}$ and $Z$, which then yields

$$\begin{aligned}
\mathbb{E}_{(Z,\hat{Z})}[|Z - \hat{Z}|] &= \mathbb{E}_Z\left[|Z - \operatorname{clip}_{\ell,r}(Z)|\right] \\
&= \mathbb{E}_Z\left[[Z - r]_+ + [\ell - Z]_+\right] \\
&= \mathbb{E}_Z\left[[Z - e^\varepsilon \cdot \mu]_+\right] + \mathbb{E}_Z\left[\left[e^{-\varepsilon} \cdot \mu - Z\right]_+\right]
\end{aligned}$$

$$= \mathbb{E}_Z\left[\left[Z - e^\varepsilon \cdot \mathbb{E}_{\tilde{Z}}[\tilde{Z}]\right]_+\right] + \mathbb{E}_Z\left[\left[e^{-\varepsilon} \cdot \mathbb{E}_{\tilde{Z}}[\tilde{Z}] - Z\right]_+\right]$$

$$\leq \mathbb{E}_{Z,\tilde{Z}}\left[\left[Z - e^\varepsilon \cdot \tilde{Z}\right]_+\right] + \mathbb{E}_{Z,\tilde{Z}}\left[\left[e^{-\varepsilon} \cdot \tilde{Z} - Z\right]_+\right]$$

$$= (1 + e^{-\varepsilon}) \cdot \mathbb{E}_{Z,\tilde{Z}}\left[\left[Z - e^\varepsilon \tilde{Z}\right]_+\right], \tag{2}$$

where the inequality follows from the convexity of $[\cdot]_+$.

Define $\hat{\mu} := \mathbb{E}_{\hat{Z}}[\hat{Z}]$. Note that we have

$$|\hat{\mu} - \mu| \leq \mathbb{E}_{(Z,\hat{Z})}[|Z - \hat{Z}|]. \tag{3}$$

Next, consider two cases based on $\mu, \mu^*$.

- Case I: $\mu > e^\varepsilon \cdot \mu^*$ or $\mu < e^{-\varepsilon} \cdot \mu^*$. We assume w.l.o.g. that $\mu > e^\varepsilon \cdot \mu^*$; the case $\mu < e^{-\varepsilon} \cdot \mu^*$ can be argued similarly. In this case, simply set $Z' = \mu^*$ always. Obviously, items (i) and (ii) hold. Moreover, we have

$$\mathbb{E}_{(Z,Z')}[|Z - Z'|] = \mathbb{E}_Z[|Z - \mu^*|] \leq \mathbb{E}_{(Z,\hat{Z})}[|\hat{Z} - \mu^*| + |\hat{Z} - Z|]$$

$$\overset{(\star)}{=} \mathbb{E}_{\hat{Z}}[\hat{Z} - \mu^*] + \mathbb{E}_{(Z,\hat{Z})}[|\hat{Z} - Z|]$$

$$= \hat{\mu} - \mu^* + \mathbb{E}_{(Z,\hat{Z})}[|\hat{Z} - Z|]$$

$$\overset{(3)}{\leq} \mu - \mu^* + 2\mathbb{E}_{(Z,\hat{Z})}[|\hat{Z} - Z|],$$

  where $(\star)$ follows from $\hat{Z} \geq e^{-\varepsilon}\mu \geq \mu^*$. Combining the above inequality with (2) yields the desired bound in item (iii).

- Case II: $\mu^* \approx_\varepsilon \mu$. In this case, we already have $\mathrm{supp}(\hat{Z}) \subseteq [e^{-\varepsilon} \cdot \mu, e^\varepsilon \cdot \mu] \subseteq [e^{-2\varepsilon}\mu^*, e^{2\varepsilon}\mu^*]$. We assume w.l.o.g. that $\hat{\mu} \geq \mu^*$; the case $\hat{\mu} \leq \mu^*$ can be handled similarly. In this case, let $f : [\ell, r] \to \mathbb{R}$ be defined by $f(\tau) := \mathbb{E}_{\hat{Z}}[\mathrm{clip}_{\ell,\tau}(\hat{Z})]$. Notice that $f$ is continuous, $f(\ell) = \ell \leq \mu^*$ and $f(r) = \hat{\mu} \geq \mu^*$. Thus, there exists $\tau^*$ such that $f(\tau^*) = \mu^*$. We then define $Z'$ by $Z' := \mathrm{clip}_{\ell,\tau^*}(\hat{Z})$. This immediately satisfies items (i) and (ii). Furthermore, this gives a natural coupling between $Z', \hat{Z}$ (and also $Z$) such that

$$\mathbb{E}_{(\hat{Z},Z')}[|\hat{Z} - Z'|] = \mathbb{E}_{(\hat{Z},Z')}[\hat{Z} - Z'] = \hat{\mu} - \mu^* \leq |\mu - \mu^*| + |\hat{\mu} - \mu|.$$

  Combining this with (2) and (3) yields the desired bound in item (iii). □

**Concentration for Bounded-Ratio Martingales.** Since we will be applying Lemma 27 on the logarithm of the ratios of random variables, it is crucial to bound its expectation. We present a bound below. Note that similar statements have been used before in DP literature; we present the (simple) proof here for completeness.

**Lemma 34.** *Let $Z$ be any $\mathbb{R}_{>0}$-valued random variable such that $\mathbb{E}[Z] = 1$. For any sufficiently small $\varepsilon > 0$, if $Z \approx_\varepsilon 1$, then $-8\varepsilon^2 \leq \mathbb{E}[\ln Z] \leq 0$.*

*Proof.* By the concavity of $\ln(\cdot)$, we get $\mathbb{E}[\ln Z] \leq \ln \mathbb{E}[Z] = 0$.

Note also that for $\varepsilon \leq 1$, we have $\ln(x) \geq x - 1 - 2(x-1)^2$ for all $x \in [e^{-\varepsilon}, e^\varepsilon]$. This implies that

$$\mathbb{E}[\ln Z] \geq \mathbb{E}[Z - 1 - 2(Z-1)^2] = -2\mathbb{E}[(Z-1)^2] \geq -2(e^\varepsilon - 1)^2 \geq -8\varepsilon^2. \qquad \square$$

We are now ready to give a concentration inequality for martingales with bounded ratios. Note that the lemma below is stated in an "expected deviation from $[e^{-\varepsilon'} \cdot \mu, e^{\varepsilon'} \cdot \mu]$" form since it will be more convenient to use in a subsequent step.

**Lemma 35.** *Assume that $\varepsilon, \delta > 0$, and $\varepsilon\sqrt{n\log(1/\delta)} > 0$ are sufficiently small. Let $Y_1, \ldots, Y_n$ be a $\mathbb{R}_{\geq 0}$-valued martingale such that $Y_i \approx_\varepsilon Y_{i-1}$ almost surely, and $\mu := \mathbb{E}[Y_1]$. Then, we have*

$$\mathbb{E}[[Y_n - e^{\varepsilon'} \cdot \mu]_+ + [\mu - e^{\varepsilon'} \cdot Y_n]_+] \leq O(\mu \cdot \delta),$$

*where $\varepsilon' = O(\varepsilon\sqrt{n\log(1/\delta)})$.*

*Proof.* Let $\tau = 100 \log(1/\delta)$. Consider $C_1, \ldots, C_n$ defined by $C_i := \ln(Y_i/Y_{i-1})$, where we use the convention $Y_0 := \mu$. By our assumption, we have $|C_i| \leq \varepsilon$. Furthermore, Lemma 34 implies that $-8\varepsilon^2 \leq \mathbb{E}[Y_i \mid Y_{i-1} = y_{i-1}, \ldots, Y_1 = y_1] \leq 0$ for all $y_{i-1}, \ldots, y_1$. Let $\gamma := -8\varepsilon^2$. Applying Lemma 27, the following holds for all $G > 0$:

$$\Pr\left[ n\gamma - \varepsilon\sqrt{nG} \leq \sum_{i \in [k]} C_i \leq \varepsilon\sqrt{nG} \right] \geq 1 - 2\exp\left(-G/2\right).$$

This is equivalent to

$$\Pr\left[ e^{n\gamma - \varepsilon\sqrt{nG}} \cdot \mu \leq Y_k \leq e^{\varepsilon\sqrt{nG}} \cdot \mu \right] \geq 1 - 2\exp\left(-G/2\right). \tag{4}$$

Thus, we have

$$\mathbb{E}\left[ [\mu - e^{\varepsilon\sqrt{n\cdot\tau}} \cdot Y_n]_+ \right] \leq \mu \cdot \Pr[Y_n \leq e^{-\varepsilon\sqrt{n\cdot\tau}} \cdot \mu] \leq \mu \cdot \Pr[Y_n \leq e^{n\gamma - \varepsilon\sqrt{n\cdot\tau}/2}\mu] \overset{(4)}{\leq} \mu \cdot \delta,$$

where the second inequality follows from the assumption that $\varepsilon\sqrt{n \log(1/\delta)}$ is sufficiently small.

Furthermore, we have

$$\mathbb{E}\left[ [Y_n - e^{\varepsilon\sqrt{n\cdot\tau}} \cdot \mu]_+ \right] = \int_0^\infty \Pr[Y_n - e^{\varepsilon\sqrt{n\cdot\tau}} \cdot \mu \geq y]dy$$

$$= \mu \cdot \int_0^\infty \Pr[Y_n - e^{\varepsilon\sqrt{n\cdot\tau}} \cdot \mu \geq \mu \cdot y]dy$$

$$= \mu \cdot \int_{e^{\varepsilon\sqrt{n\cdot\tau}}}^\infty \Pr[Y_n \geq \mu \cdot y] \, dy$$

$$= \mu \cdot \int_\tau^\infty \Pr[Y_n \geq e^{\varepsilon\sqrt{nG}} \cdot \mu] \cdot \frac{\sqrt{n}\varepsilon}{2\sqrt{G}} \exp(\varepsilon\sqrt{Gn}) \, dG$$

$$\overset{(4)}{\leq} \mu \cdot \int_\tau^\infty 2\exp(-G/2) \cdot \frac{\sqrt{n}\varepsilon}{2\sqrt{G}} \exp(\varepsilon\sqrt{Gn}) \, dG$$

$$\leq \mu \cdot \int_\tau^\infty 2\exp(-G/4) \, dG$$

$$= \mu \cdot 8\exp(-\tau/4)$$

$$\leq \mu \cdot \delta,$$

where the second and third inequalities are from our choice of $\tau$.

Combining the previous two inequalities yields the desired bound for $\varepsilon' = \varepsilon\sqrt{n\tau}$. $\qquad \square$

**Putting Things Together: Proof of Lemma 32.** Our overall strategy is similar to the proof for the always-bounded case, which is to set up a martingale from suffix averages and then apply Lemma 35. The main modification is that, instead of using this argument directly on $f$ (and its suffix averages), we use it on a different function, which is the result of applying Lemma 33 to $f$.

*Proof of Lemma 32.* For convenience, let $\mu := \mu_{\mathcal{D}^n}(f)$. Let us first extend the function $f$ to include prefixes (i.e., $f : \mathcal{Z}^{\leq n} \to \mathbb{R}_{\geq 0}$) naturally as follows:

$$f(x_1, \ldots, x_i) = \mathbb{E}_{X_{i+1}, \ldots, X_n \sim \mathcal{D}}[f(x_1, \ldots, x_i, X_{i+1}, \ldots, X_n)].$$

Next, we construct a function $g : \mathcal{Z}^{\leq n} \to \mathbb{R}_{\geq 0}$ as follows:

- $g(\emptyset) = f(\emptyset)$ (which is equal to $\mu$)
- For $i = 1, \ldots, n$:
  - For all $(x_1, \ldots, x_{i-1}) \in \mathcal{Z}^{i-1}$:
    * Apply Lemma 33 with $\mu^* := g(x_1, \ldots, x_{i-1})$ and $Z = f(x_1, \ldots, x_{i-1}, X_i)$, where $X_i \sim \mathcal{D}$ to get a random variable $Z'$.
    * The coupling between $(Z, Z')$ is naturally also a coupling between $(x_i, Z')$ for $x_i \in \mathcal{Z}$. Let $g(x_1, \ldots, x_i) = Z'$ based on this coupling for all $x_i \in \mathcal{Z}$.

For brevity, let $\rho = 2(1 + e^{-\varepsilon})$. We have

$$\mathbb{E}_{\boldsymbol{x} \sim \mathcal{D}^n}[|g(\boldsymbol{x}) - f(\boldsymbol{x})|]$$

$$= \mathbb{E}_{\boldsymbol{x} \sim \mathcal{D}^n} \mathbb{E}_{\tilde{x}_n \sim \mathcal{D}^n}[|g(\boldsymbol{x}_{\leq n-1}, \tilde{x}_n) - f(\boldsymbol{x}_{\leq n-1}, \tilde{x}_n)|]$$

$$\leq \mathbb{E}_{\boldsymbol{x} \sim \mathcal{D}^n}\left[|g(\boldsymbol{x}_{\leq n-1}) - f(\boldsymbol{x}_{\leq n-1})| + \rho \cdot \mathbb{E}_{\tilde{x}_n, \tilde{x}'_n \sim \mathcal{D}}[f(\boldsymbol{x}_{\leq n-1}, \tilde{x}_n) - e^{\varepsilon} \cdot f(\boldsymbol{x}_{\leq n-1}, \tilde{x}'_n)]_+\right]$$

$$= \mathbb{E}_{\boldsymbol{x}_{\leq n-1} \sim \mathcal{D}^{n-1}}[|g(\boldsymbol{x}_{\leq n-1}) - f(\boldsymbol{x}_{\leq n-1})|] + \rho \cdot \mathbb{E}_{\boldsymbol{x}, \boldsymbol{x}' \sim \mathcal{D}^n}[f(\boldsymbol{x}_{\leq n-1}, x_n) - e^{\varepsilon} \cdot f(\boldsymbol{x}_{\leq n-1}, x'_n)]_+,$$

where the inequality is from Lemma 33(iii).

By repeatedly applying the above argument (to the first term), we arrive at

$$\mathbb{E}_{\boldsymbol{x} \sim \mathcal{D}^n}[|g(\boldsymbol{x}) - f(\boldsymbol{x})|] \leq \rho \cdot \sum_{i=1}^{n} \mathbb{E}_{\boldsymbol{x}, \boldsymbol{x}' \sim \mathcal{D}^n}[f(\boldsymbol{x}_{\leq i-1}, x_i) - e^{\varepsilon} \cdot f(\boldsymbol{x}_{\leq i-1}, x'_i)]_+.$$

Recall from the definition that $f(\boldsymbol{x}_{\leq i-1}, x_i) = \mathbb{E}_{\boldsymbol{x}_{>i} \sim \mathcal{D}^{n-i}}[f(\boldsymbol{x}_{\leq i-1}, x_i, \boldsymbol{x}_{>i})]$ and $f(\boldsymbol{x}_{\leq i-1}, x'_i) = \mathbb{E}_{\boldsymbol{x}_{>i} \sim \mathcal{D}^{n-i}}[f(\boldsymbol{x}_{\leq i-1}, x'_i, \boldsymbol{x}_{>i})]$. Thus, by the convexity of $[\cdot]_+$, we further have

$$\mathbb{E}_{\boldsymbol{x} \sim \mathcal{D}^n}[|g(\boldsymbol{x}) - f(\boldsymbol{x})|] \leq \rho \cdot \sum_{i=1}^{n} \mathbb{E}_{\boldsymbol{x}, \tilde{\boldsymbol{x}} \sim \mathcal{D}^n}\left[f(\boldsymbol{x}) - e^{\varepsilon} \cdot f(\tilde{\boldsymbol{x}}^{(i)})\right]_+. \tag{5}$$

Next, let $Y_0, Y_1, \ldots, Y_n$ be a sequence of random variables defined by sampling $x_1, \ldots, x_n \sim \mathcal{D}^n$ and letting $Y_i := g(x_1, \ldots, x_i)$. By Lemma 33(ii), we have that $Y_1, \ldots, Y_n$ is a martingale. Lemma 33(i) further implies that $Y_{i-1} \approx_{2\varepsilon} Y_i$. Thus, we may apply Lemma 35 to arrive at

$$\mathbb{E}[[Y_n - e^{\varepsilon'/2} \cdot \mu]_+ + [\mu - e^{\varepsilon'/2} \cdot Y_n]_+] \leq O(\mu\delta),$$

for $\varepsilon' = O(\varepsilon\sqrt{n\log(1/\delta)})$. Note that this is equivalent to

$$\mathbb{E}_{\boldsymbol{x} \sim \mathcal{D}^n}[[g(\boldsymbol{x}) - e^{\varepsilon'/2} \cdot \mu]_+ + [\mu - e^{\varepsilon'/2} \cdot g(\boldsymbol{x})]_+] \leq O(\mu\delta). \tag{6}$$

We can now derive the bound on the desired quantity as follows:

$$\mathbb{E}_{\boldsymbol{x}, \boldsymbol{x}' \sim \mathcal{D}^n}\left[[f(\boldsymbol{x}) - e^{\varepsilon'} f(\boldsymbol{x}')]_+\right]$$

$$= \mathbb{E}_{\boldsymbol{x}, \boldsymbol{x}' \sim \mathcal{D}^n}\left[[(f(\boldsymbol{x}) - g(\boldsymbol{x})) + (g(\boldsymbol{x}) - e^{\varepsilon'/2}\mu) + e^{\varepsilon'/2}(\mu - e^{\varepsilon'/2}g(\boldsymbol{x}')) + e^{\varepsilon'}(g(\boldsymbol{x}') - f(\boldsymbol{x}'))]_+\right]$$

$$\leq \mathbb{E}[f(\boldsymbol{x}) - g(\boldsymbol{x})]_+ + \mathbb{E}[g(\boldsymbol{x}) - e^{\varepsilon'/2}\mu]_+ + e^{\varepsilon'/2}\mathbb{E}[\mu - e^{\varepsilon'/2}g(\boldsymbol{x}')]_+ + \mathbb{E}[g(\boldsymbol{x}') - f(\boldsymbol{x}')]_+$$

$$\overset{(5),(6)}{\leq} O(\mu\delta) + O\left(\sum_{i=1}^{n} \mathbb{E}_{\boldsymbol{x}, \tilde{\boldsymbol{x}} \sim \mathcal{D}^n}\left[f(\boldsymbol{x}) - e^{\varepsilon} \cdot f(\tilde{\boldsymbol{x}}^{(i)})\right]_+\right). \qquad \square$$

### C.3.2 Putting Things Together: Proof of Lemma 14

Lemma 14 is now a simple consequence of Lemma 32.

*Proof of Lemma 14.* For every $o \in \mathcal{O}$ and $\boldsymbol{x} \in \mathcal{Z}^n$, let $f^o(\boldsymbol{x}) := \Pr[\mathbb{A}(\boldsymbol{x}) = o]$. Applying Lemma 32 to $f^o$, we get

$$\mathbb{E}_{\boldsymbol{x}, \boldsymbol{x}' \sim \mathcal{D}^n}[[f^o(\boldsymbol{x}) - e^{\varepsilon'} \cdot f^o(\boldsymbol{x}')]_+]$$

$$\leq O(\mu_{\mathcal{D}^n}(f^o) \cdot \delta) + O\left(\sum_{i=1}^{n} \mathbb{E}_{\boldsymbol{x}, \tilde{\boldsymbol{x}} \sim \mathcal{D}^n}\left[f^o(\boldsymbol{x}) - e^{\varepsilon} \cdot f^o(\tilde{\boldsymbol{x}}^{(i)})\right]_+\right),$$

where $\varepsilon' = O(\varepsilon\sqrt{n\log(1/\delta)})$.

Summing this up over all $o \in \mathcal{O}$, we get

$$\mathbb{E}_{\boldsymbol{x}, \boldsymbol{x}' \sim \mathcal{D}^n}[d_{\varepsilon'}(\mathbb{A}(\boldsymbol{x}) \parallel \mathbb{A}(\boldsymbol{x}'))] \leq O(\delta) + O\left(\sum_{i=1}^{n} \mathbb{E}_{\boldsymbol{x}, \tilde{\boldsymbol{x}} \sim \mathcal{D}^n}\left[d_{\varepsilon}(\mathbb{A}(\boldsymbol{x}) \parallel \mathbb{A}(\tilde{\boldsymbol{x}}^{(i)}))\right]\right) \leq O(n\delta),$$

where the second inequality follows from the assumption that $\mathbb{A}$ is $(\varepsilon, \delta)$-item-level DP, which implies $d_{\varepsilon}(\mathbb{A}(\boldsymbol{x}) \parallel \mathbb{A}(\tilde{\boldsymbol{x}}^{(i)})) \leq \delta$. $\qquad \square$

## C.4 Privacy Analysis of Algorithm 1

Here we prove the privacy guarantee (Lemma 18) of Algorithm 1. This mostly follows the proof for the analogous algorithm in [GKK+23].

We start with the following observation.

**Observation 36.** *For user-level neighboring datasets $x, x'$, and all $r_1 \in \{0, \ldots, n-1\}$, if $\mathcal{X}_{\text{stable}}^{r_1}(x') \neq \emptyset$, then $\mathcal{X}_{\text{stable}}^{r_1+1}(x) \neq \emptyset$.*

*Proof.* Let $i \in [n]$ be such that $x_{-i} = x'_{-i}$, and let $x'_{-S'}$ be an element of $\mathcal{X}_{\text{stable}}^{r_1}(x')$. Select $S$ such that $|S| = r_1 + 1$ and $S \supseteq S' \cup \{i\}$. It is obvious from definition that $x_S \in \mathcal{X}_{\text{stable}}^{r_1+1}(x)$. $\square$

*Proof of Lemma 18.* Let $x, x'$ be user-level neighboring datasets, and let $\mathcal{A}$ be a shorthand for $\mathsf{DelStab}_{\varepsilon,\delta,\mathbb{A}}$. First, we follow the proof of [GKK+23, Theorem 3.3]:

$$\begin{aligned}
\Pr[\mathcal{A}(x) = \perp] &= \sum_{r_1=0}^{2\kappa} \mathbf{1}[\mathcal{A}(x) = \perp \mid R_1 = r_1] \cdot \Pr[R_1 = r_1] \\
&= \sum_{r_1=0}^{2\kappa} \mathbf{1}[\mathcal{X}_{\text{stable}}^{r_1}(x) = \emptyset] \cdot \Pr[R_1 = r_1] \\
&\leq \Pr[R_1 = 0] + \sum_{r_1=1}^{2\kappa} \mathbf{1}[\mathcal{X}_{\text{stable}}^{r_1}(x) = \emptyset] \cdot \Pr[R_1 = r_1] \\
&\leq \overline{\delta} + \sum_{r_1=1}^{2\kappa} \mathbf{1}[\mathcal{X}_{\text{stable}}^{r_1}(x) = \emptyset] \cdot e^{\overline{\varepsilon}} \cdot \Pr[R_1 = r_1 - 1] \\
&\leq \overline{\delta} + \sum_{r_1=1}^{2\kappa} \mathbf{1}[\mathcal{X}_{\text{stable}}^{r_1-1}(x') = \emptyset] \cdot e^{\overline{\varepsilon}} \cdot \Pr[R_1 = r_1 - 1] \\
&\leq \overline{\delta} + e^{\overline{\varepsilon}} \cdot \sum_{r_1=0}^{2\kappa} \mathbf{1}[\mathcal{A}(x') = \perp \mid R_1 = r_1] \cdot \Pr[R_1 = r_1] \\
&= \overline{\delta} + e^{\overline{\varepsilon}} \cdot \Pr[\mathcal{A}(x') = \perp], \tag{7}
\end{aligned}$$

and for any $U_0 \subseteq \mathcal{O}$, we have

$$\begin{aligned}
\Pr[\mathcal{A}(x) \in U_0] &\leq \sum_{r_1=0}^{2\kappa} \Pr[\mathcal{A}(x) \in U_0 \mid R_1 = r_1] \cdot \Pr[R_1 = r_1] \\
&= \sum_{r_1=0}^{2\kappa-1} \Pr[\mathcal{A}(x) \in U_0 \mid R_1 = r_1] \cdot \Pr[R_1 = r_1] + \Pr[R_1 = 2\kappa] \\
&\leq \overline{\delta} + e^{\overline{\varepsilon}} \cdot \sum_{r_1=0}^{2\kappa-1} \Pr[\mathcal{A}(x) \in U_0 \mid R_1 = r_1] \cdot \Pr[R_1 = r_1 + 1], \tag{8}
\end{aligned}$$

To bound the term $\Pr[\mathcal{A}(x) \in U_0 \mid R_1 = r_1]$ for $r_1 < 2\kappa$, observe that if it is non-zero, then it must be that $\mathcal{A}(x) \neq \perp$ or equivalently that $\mathcal{X}_{\text{stable}}^{r_1}(x) \neq \emptyset$. In this case, we have[10]

$$\Pr[\mathcal{A}(x) \in U_0 \mid R_1 = r_1] = \mathbb{E}_{S \sim \mathcal{X}_{\text{stable}}^{r_1}(x), J \sim \binom{\overline{S}}{n-4\kappa}} [\Pr[\mathbb{A}(x_J) \in U_0]].$$

Since $\mathcal{X}_{\text{stable}}^{r_1}(x) \neq \emptyset$, Observation 36 also implies that[11] $\mathcal{X}_{\text{stable}}^{r_1+1}(x') \neq \emptyset$. Thus, we have

$$\Pr[\mathcal{A}(x') \in U_0 \mid R_1 = r_1 + 1] = \mathbb{E}_{S' \sim \mathcal{X}_{\text{stable}}^{r_1+1}(x'), J' \sim \binom{\overline{S}'}{n-4\kappa}} [\Pr[\mathbb{A}(x_{J'}) \in U_0]].$$

Consider an arbitrary pair $S \in \mathcal{X}_{\text{stable}}^{r_1}(x), S' \in \mathcal{X}_{\text{stable}}^{r_1+1}(x')$. Let $i \in [n]$ be such that $x_{-i} = x'_{-i}$. Note that $|(\overline{S} \cap \overline{S}') \smallsetminus \{i\}| \geq n - 4\kappa$. Let $J^*$ be any subset of $(\overline{S} \cap \overline{S}')$ of size $n - 4\kappa$. From $S \in \mathcal{X}_{\text{stable}}^{r_1}(x)$, we have

$$\Pr[\mathbb{A}(x_J) \in U_0] \leq e^{\overline{\varepsilon}} \Pr[\mathbb{A}(x_{J^*}) \in U_0] + \overline{\delta},$$

for all $J \in \binom{\overline{S}}{n-4\kappa}$, and, from $S' \in \mathcal{X}_{\text{stable}}^{r_1+1}(x')$, we have

$$\Pr[\mathbb{A}(x_{J^*}) \in U_0] \leq e^{\overline{\varepsilon}} \Pr[\mathbb{A}(x_{J'}) \in U_0] + \overline{\delta},$$

for all $J' \in \binom{\overline{S}'}{n-4\kappa}$.

Therefore, by arbitrary coupling of $S, S'$, we can conclude that

$$\Pr[\mathcal{A}(x) \in U_0 \mid R_1 = r_1] \leq e^{2\overline{\varepsilon}} \Pr[\mathcal{A}(x') \in U_0 \mid R_1 = r_1 + 1] + (e^{\overline{\varepsilon}} + 1)\overline{\delta}.$$

---

[10]Here we write $\overline{S} := [n] \smallsetminus S$.
[11]Again we write $\overline{S}' := [n] \smallsetminus S'$.

Plugging this back to (8), we get

$$\Pr[\mathcal{A}(\boldsymbol{x}) \in U_0]$$

$$\leq \overline{\delta} + e^{\overline{\varepsilon}} \cdot \sum_{r_1=0}^{2\kappa-1} \left(e^{2\overline{\varepsilon}} \Pr[\mathcal{A}(\boldsymbol{x}') \in U_0 \mid R_1 = r_1 + 1] + (e^{\overline{\varepsilon}} + 1)\overline{\delta}\right) \cdot \Pr[R_1 = r_1 + 1]$$

$$\leq (e^{2\overline{\varepsilon}} + e^{\overline{\varepsilon}} + 1)\overline{\delta} + e^{3\overline{\varepsilon}} \cdot \sum_{r_1=0}^{2\kappa} \Pr[\mathcal{A}(\boldsymbol{x}') \in U_0 \mid R_1 = r_1] \cdot \Pr[R_1 = r_1]$$

$$= (e^{2\overline{\varepsilon}} + e^{\overline{\varepsilon}} + 1)\overline{\delta} + e^{3\overline{\varepsilon}} \cdot \Pr[\mathbb{A}(\boldsymbol{x}') \in U_0]. \tag{9}$$

Now, consider any set $U \subseteq \mathcal{O} \cup \{\bot\}$ of outcomes. Let $U_{\mathcal{O}} = U \cap \mathcal{O}$ and $U_{\bot} = U \cap \{\bot\}$. Then, we have

$$\Pr[\mathcal{A}(\boldsymbol{x}) \in U] = \Pr[\mathcal{A}(\boldsymbol{x}) \in U_{\mathcal{O}}] + \Pr[\mathcal{A}(\boldsymbol{x}) \in U_{\bot}]$$

$$\overset{(9),(7)}{\leq} \left((e^{2\overline{\varepsilon}} + e^{\overline{\varepsilon}} + 1)\overline{\delta} + e^{3\overline{\varepsilon}} \cdot \Pr[\mathbb{A}(\boldsymbol{x}') \in U_{\mathcal{O}}]\right) + \left(\overline{\delta} + e^{\overline{\varepsilon}} \cdot \Pr[\mathcal{A}(\boldsymbol{x}') = U_{\bot}]\right)$$

$$\leq (e^{2\overline{\varepsilon}} + e^{\overline{\varepsilon}} + 2)\overline{\delta} + e^{3\overline{\varepsilon}} \Pr[\mathcal{A}(\boldsymbol{x}') \in U]$$

$$\leq \delta + e^{\varepsilon} \cdot \Pr[\mathcal{A}(\boldsymbol{x}') \in U].$$

Therefore, the algorithm is $(\varepsilon, \delta)$-user-level DP as desired. $\qquad\square$

## D    Missing Details from Section 4

Below we give the proof of Lemma 22.

*Proof of Lemma 22.* Consider any $h \in \{0,1\}^{\mathcal{X}}$. Since $\boldsymbol{z}_i$'s are drawn independently from $\mathcal{D}$, we have that $\mathbf{1}[\boldsymbol{z}_i$ is not realizable by $h]$'s are independent Bernoulli random variables with success probability

$$p_h = 1 - (1 - \mathrm{err}_{\mathcal{D}}(h))^m.$$

Consider the two cases:

- If $\mathrm{err}_{\mathcal{D}}(h) \leq 0.01\alpha$, then we have

$$p_h \leq 1 - (1 - 0.01\alpha)^m \leq 0.01\alpha m.$$

  Thus, applying Lemma 25, we can conclude that

$$\Pr[\mathrm{scr}_h(\boldsymbol{z}) \geq 0.05\alpha nm] \leq \exp\left(-\Theta\left(n\alpha m\right)\right) = \exp\left(-\Theta\left(\kappa \cdot \mathrm{size}(\mathcal{P})\right)\right).$$

  Thus, when we select $\kappa$ to be sufficiently large, we have

$$\Pr[\mathrm{scr}_h(\boldsymbol{z}) \geq 0.05\alpha nm] \leq \exp(-10\,\mathrm{size}(\mathcal{P})) \leq \frac{0.01}{\exp(\mathrm{size}(\mathcal{P}))} \leq \frac{0.01}{|\mathcal{H}|}.$$

- If $\mathrm{err}_{\mathcal{D}}(h) > \alpha$, then we have

$$p_h \geq 1 - (1 - \alpha)^m \geq 1 - e^{-\alpha m} \geq 0.2\alpha m,$$

  where in the second inequality we use the fact that $\alpha m \leq 1$.
  Again, applying Lemma 25, we can conclude that

$$\Pr[\mathrm{scr}_h(\boldsymbol{z}) \leq 0.2\alpha nm] \leq \exp\left(-\Theta\left(n\alpha m\right)\right) = \exp\left(-\Theta\left(\kappa \cdot \mathrm{size}(\mathcal{P})\right)\right).$$

  Thus, when we select $\kappa$ to be sufficiently large, we have

$$\Pr[\mathrm{scr}_h(\boldsymbol{z}) \leq 0.2\alpha nm] \leq \exp(-10\,\mathrm{size}(\mathcal{P})) \leq \frac{0.01}{\exp(\mathrm{size}(\mathcal{P}))} \leq \frac{0.01}{|\mathcal{H}|}.$$

Finally, taking a union bound over all $h \in \mathcal{H}$ yields the desired result. $\qquad\square$

It is also worth noting that the scoring function can be written in the form:

$$\mathrm{scr}_h(\boldsymbol{z}) := \sum_{i \in [n]} \mathrm{clip}_{0,1}\left(|\{(x, y) \in \boldsymbol{z}_i \mid h(x) \neq y\}|\right),$$

which is similar to what we present below for other tasks.

Finally, we remark that the standard techniques can boost the success probability from 2/3 to an arbitrary probability $1 - \beta$, with a multiplicative factor of $\log(1/\beta)$ cost.

# E  Additional Results for Pure-DP

## E.1  Algorithm: Pairwise Score Exponential Mechanism

In this section, we give a user-level DP version of *pairwise score* exponential mechanism (EM) [BKSW19]. In the proceeding subsections, we will show how to apply this in multiple settings, including agnostic learning, private hypothesis selection, and distribution learning.

Suppose there is a candidate set $\mathcal{H}$. Furthermore, for every pair $H, H' \in \mathcal{H}$ of candidates, there is a "comparison" function $\psi_{H,H'} : \mathcal{Z} \to [-1, 1]$. For every pair $H, H'$ of candidates and any $\boldsymbol{x} \in \mathcal{Z}^*$, let

$$\mathrm{pscr}_{H,H'}^{\psi}(\boldsymbol{x}) = \sum_{x \in \boldsymbol{x}} \psi_{H,H'}(x).$$

We then define $\mathrm{scr}^{\psi}$ for each candidate $H$ as

$$\mathrm{scr}_H^{\psi}(\boldsymbol{x}) := \max_{H' \in \mathcal{H} \smallsetminus \{H\}} \mathrm{pscr}_{H,H'}^{\psi}(\boldsymbol{x}).$$

Traditionally, the item-level DP algorithm for pairwise scoring function is to use EM on the above scoring function $\mathrm{scr}_H^{\psi}$ (where $\boldsymbol{x}$ denote the entire input) [BKSW19]; it is not hard to see that the $\mathrm{scr}_H^{\psi}$ has (item-level) sensitivity of at most 2.

**User-Level DP Algorithm.** We now describe our user-level DP algorithm ClippedPairwiseEM. The algorithm computes:

$$\mathrm{pscr}_{H,H'}^{\psi,\tau}(\boldsymbol{x}) := \sum_{i \in [n]} \mathrm{clip}_{-\tau,\tau}\left(\mathrm{pscr}_{H,H'}^{\psi}(\boldsymbol{x}_i)\right),$$

$$\mathrm{scr}_H^{\psi,\tau}(\boldsymbol{x}) := \max_{H' \in \mathcal{H} \smallsetminus \{H\}} \mathrm{pscr}_{H,H'}^{\psi,\tau}(\boldsymbol{x}).$$

Then, it simply runs EM using the score $\mathrm{scr}_H^{\psi,\tau}$ with sensitivity $\Delta = 2\tau$.

To state the guarantee of the algorithm, we need several additional distributional-based definitions of the score:

$$\mathrm{pscr}_{H,H'}^{\psi}(\mathcal{D}) := \mathbb{E}_{x \sim \mathcal{D}}\left[\psi_{H,H'}(x)\right],$$

$$\mathrm{scr}_H^{\psi}(\mathcal{D}) := \max_{H' \in \mathcal{H} \smallsetminus \{H\}} \mathrm{pscr}_{H,H'}^{\psi}(\mathcal{D}).$$

Note that these distributional scores are normalized, so they are in $[-1, 1]$.

The guarantee of our algorithm can now be stated as follows:

**Theorem 37.** *Let $\varepsilon, \alpha, \beta \in (0, 1/2]$. Assume further that there exists $H^*$ such that $\mathrm{scr}_{H^*}^{\psi}(\mathcal{D}) \leq 0.1\alpha$. There exists an $\varepsilon$-DP algorithm that with probability $1 - \beta$ (over the randomness of the algorithm and the input $\boldsymbol{x} \sim (\mathcal{D}^m)^n$) outputs $H \in \mathcal{H}$ such that $\mathrm{scr}_H^{\psi}(\mathcal{D}) \leq 0.5\alpha$ as long as*

$$n \geq \tilde{\Theta}\left(\frac{\log(|\mathcal{H}|/\beta)}{\varepsilon} + \frac{\log(|\mathcal{H}|/\beta)}{\alpha\varepsilon\sqrt{m}} + \frac{\log(|\mathcal{H}|/\beta)}{\alpha^2 m}\right).$$

*Proof.* We use ClippedPairwiseEM with parameter $\tau = C\left(\alpha m + \sqrt{m \log(1/\alpha)}\right)$ and assume that $n \geq C'\left(\frac{\tau \cdot \log(|\mathcal{H}|/\beta)}{\varepsilon\alpha m} + \frac{\log(|\mathcal{H}|/\beta)}{\alpha^2 m}\right) = \tilde{\Theta}\left(\frac{\log(|\mathcal{H}|/\beta)}{\varepsilon} + \frac{\log(|\mathcal{H}|/\beta)}{\alpha\varepsilon\sqrt{m}} + \frac{\log(|\mathcal{H}|/\beta)}{\alpha^2 m}\right)$, where $C, C'$ are sufficiently large constants. The privacy guarantee follows directly from that of EM, since the (user-level) sensitivity is at most $\Delta$ (due to clipping).

To analyze the utility, we will consider following three events:

- $\mathcal{E}_1$: $\mathrm{scr}_H^{\psi,\tau}(\boldsymbol{x}) \leq \mathrm{scr}_{H^*}^{\psi,\tau}(\boldsymbol{x}) + 0.1\alpha nm$.
- $\mathcal{E}_2$: for all $H, H' \in \mathcal{H}$: if $\mathrm{pscr}_{H,H'}^{\psi}(\mathcal{D}) \leq 0.1\alpha$, then $\mathrm{pscr}_{H,H'}^{\psi}(\boldsymbol{x}) \leq 0.2\alpha nm$.
- $\mathcal{E}_3$: for all $H, H' \in \mathcal{H}$: if $\mathrm{pscr}_{H,H'}^{\psi}(\mathcal{D}) \geq 0.5\alpha$, then $\mathrm{pscr}_{H,H'}^{\psi}(\boldsymbol{x}) \geq 0.4\alpha nm$.

When all events hold, $\mathcal{E}_1$ implies that $\mathrm{scr}_H^{\psi,\tau}(\boldsymbol{x}) \leq \mathrm{scr}_{H^*}^{\psi,\tau}(\boldsymbol{x}) + 0.1\alpha nm$. Observe that $\mathcal{E}_2$ implies that $\mathrm{scr}_{H^*}^{\psi,\tau}(\boldsymbol{x}) \leq 0.2\alpha nm$. Combining these two inequalities, we get $\mathrm{scr}_H^{\psi,\tau}(\boldsymbol{x}) \leq 0.3\alpha nm$; then, $\mathcal{E}_3$ implies that $\mathrm{pscr}_{H,H'}^{\psi}(\mathcal{D}) \leq 0.5\alpha$ for all $H' \in \mathcal{D}$. In turn, this means that $\mathrm{scr}_H^{\psi}(\mathcal{D}) \leq 0.5\alpha$.

Therefore, it suffices to show that each of $\Pr[\mathcal{E}_1], \Pr[\mathcal{E}_2], \Pr[\mathcal{E}_3]$ is at least $1 - \beta/3$.

**Bounding $\Pr[\mathcal{E}_1]$.** To bound the probability of $\mathcal{E}_1$, recall from Theorem 9 that, with probability $1 - \beta/3$, the guarantee of EM ensures that

$$\mathrm{scr}_H^{\psi,\tau}(\boldsymbol{x}) \geq \mathrm{scr}_{H^*}^{\psi,\tau}(\boldsymbol{x}) + O\left(\frac{\log(|\mathcal{H}|/\beta)}{\varepsilon}\right) \cdot \Delta.$$

From our assumption on $n$, when $C'$ is sufficiently large, then we have $O\left(\frac{\log(|\mathcal{H}|/\beta)}{\varepsilon}\right) \cdot \Delta = O\left(\frac{\log(|\mathcal{H}|/\beta)}{\varepsilon} \cdot \tau\right) \leq 0.1\alpha nm$ as desired.

**Bounding $\Pr[\mathcal{E}_2]$ and $\Pr[\mathcal{E}_3]$.** Since the proofs for both cases are analogous, we only show the full argument for $\Pr[\mathcal{E}_3]$. Let us fix $H, H' \in \mathcal{H}$ such that $\mathrm{pscr}_{H,H'}^{\psi}(\mathcal{D}) \geq 0.5\alpha$. We may assume w.l.o.g. that[12] $\mathrm{pscr}_{H,H'}^{\psi}(\mathcal{D}) = 0.5\alpha := \mu$.

Notice that $\mathrm{pscr}_{H,H'}^{\psi,\tau}(\boldsymbol{x})$ is a 2-bounded difference function. As a result, McDiarmid's inequality (Lemma 26) implies that

$$\Pr_{\boldsymbol{x} \sim \mathcal{D}^{nm}}[|\mathrm{pscr}_{H,H'}^{\psi,\tau}(\boldsymbol{x}) - \mu_{\mathcal{D}^{nm}}(\mathrm{pscr}_{H,H'}^{\psi,\tau})| > 0.05\alpha nm] \leq 2\exp\left(-0.025\alpha^2 nm\right) \leq \frac{\beta}{3|\mathcal{H}|^2}, \quad (10)$$

where the second inequality is due to our choice of $n$ when $C'$ is sufficiently large.

To compute $\mu_{\mathcal{D}^{nm}}(\mathrm{pscr}_{H,H'}^{\psi,\tau})$, observe further that

$$\mu_{\mathcal{D}^{nm}}(\mathrm{pscr}_{H,H'}^{\psi,\tau}) = n \cdot \mu_{\mathcal{D}^m}(\mathrm{clip}_{-\tau,\tau} \circ \mathrm{pscr}_{H,H'}^{\psi}). \quad (11)$$

Now observe once again that $\mathrm{pscr}_{H,H'}^{\psi} : \mathcal{Z}^m \to \mathbb{R}$ is also a 2-bounded difference function and $\mu_{\mathcal{D}^m}(\mathrm{pscr}_{H,H'}^{\psi}) = \mu m$. Therefore, McDiarmid's inequality (Lemma 26) yields

$$\Pr_{\boldsymbol{x} \sim \mathcal{D}^m}[\mathrm{pscr}_{H,H'}^{\psi}(\boldsymbol{x}) > \tau] \leq 2\exp\left(-0.5(\tau - \mu m)^2/m\right) \leq 10^{-4}\alpha^2, \quad (12)$$

where the second inequality is due to our choice of $\tau$ when $C$ is sufficiently large.

Finally, note that

$$\mu_{\mathcal{D}^m}(\mathrm{clip}_{-\tau,\tau} \circ \mathrm{pscr}_{H,H'}^{\psi})$$
$$= \mathbb{E}_{\boldsymbol{x} \sim \mathcal{D}^m}[\mathrm{clip}_{-\tau,\tau}(\mathrm{pscr}_{H,H'}^{\psi}(\boldsymbol{x}))]$$
$$\geq \mathbb{E}_{\boldsymbol{x} \sim \mathcal{D}^m}[\mathrm{clip}_{-\infty,\tau}(\mathrm{pscr}_{H,H'}^{\psi}(\boldsymbol{x}))]$$
$$= \mathbb{E}_{\boldsymbol{x} \sim \mathcal{D}^m}[\mathrm{pscr}_{H,H'}^{\psi}(\boldsymbol{x})] + \mathbb{E}_{\boldsymbol{x} \sim \mathcal{D}^m}[(\tau - \mathrm{pscr}_{H,H'}^{\psi}(\boldsymbol{x})) \cdot \mathbf{1}[\mathrm{pscr}_{H,H'}^{\psi}(\boldsymbol{x}) > \tau]]$$
$$\geq \mu m - \mathbb{E}_{\boldsymbol{x} \sim \mathcal{D}^m}[\mathrm{pscr}_{H,H'}^{\psi}(\boldsymbol{x}) \cdot \mathbf{1}[\mathrm{pscr}_{H,H'}^{\psi}(\boldsymbol{x}) > \tau]]$$
$$\geq \mu m - \sqrt{\mathbb{E}_{\boldsymbol{x} \sim \mathcal{D}^m}[\mathrm{pscr}_{H,H'}^{\psi}(\boldsymbol{x})^2]} \cdot \sqrt{\mathbb{E}_{\boldsymbol{x} \sim \mathcal{D}^m}[\mathbf{1}[\mathrm{pscr}_{H,H'}^{\psi}(\boldsymbol{x}) > \tau]]}$$
$$\geq \mu m - \sqrt{m^2} \cdot \sqrt{\Pr_{\boldsymbol{x} \sim \mathcal{D}^m}[\mathrm{pscr}_{H,H'}^{\psi}(\boldsymbol{x}) > \tau]}$$
$$\overset{(12)}{\geq} \mu m - 0.01\alpha m = 0.49\alpha m, \quad (13)$$

where the third-to-last inequality follows from Cauchy–Schwarz.

---

[12]Otherwise, we may keep increasing $\psi_{H,H'}(z)$ for different values of $z$ until $\mathrm{pscr}_{H,H'}^{\psi}(\mathcal{D}) = 0.5\alpha$; this operation does not decrease the probability that $\mathrm{pscr}_{H,H'}^{\psi}(\boldsymbol{x}) < 0.4\alpha nm$.

Combining (10), (11), and (13), we can conclude that

$$\Pr_{\boldsymbol{x}\sim\mathcal{D}^{nm}}[\mathrm{pscr}_{H,H'}^{\psi,\tau}(\boldsymbol{x}) < 0.4\alpha nm] \leq \frac{\beta}{3|\mathcal{H}|^2}.$$

Taking a union bound over $H, H' \in \mathcal{H}$ yields $\Pr[\mathcal{E}_3] \geq 1 - \beta/3$ as desired. $\qquad\square$

## E.2 Lower Bound: User-level DP Fano's Inequality

As we often demonstrate below that our bounds are (nearly) tight, it will be useful to have a generic method for providing such a lower bound. Here we observe that it is simple to extend the DP Fano's inequality of Acharya et al. [ASZ21] to the user-level DP setting. We start by recalling Acharya et al.'s (item-level) DP Fano's inequality[13]:

**Theorem 38** (Item-Level DP Fano's Inequality [ASZ21]). *Let $\mathfrak{T}$ be any task. Suppose that there exist $\mathcal{D}_1, \ldots, \mathcal{D}_W$ such that, for all distinct $i, j \in [W]$,*

- $\Psi_{\mathfrak{T}}(\mathcal{D}_i) \cap \Psi_{\mathfrak{T}}(\mathcal{D}_j) = \emptyset$,
- $d_{\mathrm{KL}}(\mathcal{D}_i, \mathcal{D}_j) \leq \beta$*, and*
- $d_{\mathrm{tv}}(\mathcal{D}_i, \mathcal{D}_j) \leq \gamma$.

*Then, $n_1^{\mathfrak{T}}(\varepsilon, \delta = 0) \geq \Omega\left(\frac{\log W}{\gamma\varepsilon} + \frac{\log W}{\beta}\right)$.*

The bound for the user-level case can be derived as follows.

**Lemma 39** (User-Level DP Fano's Inequality). *Let $\mathfrak{T}$ be any task. Suppose that there exist $\mathcal{D}_1, \ldots, \mathcal{D}_W$ such that, for all distinct $i, j \in [W]$,*

- $\Psi_{\mathfrak{T}}(\mathcal{D}_i) \cap \Psi_{\mathfrak{T}}(\mathcal{D}_j) = \emptyset$*, and,*
- $d_{\mathrm{KL}}(\mathcal{D}_i, \mathcal{D}_j) \leq \beta$.

*Then, $n_m^{\mathfrak{T}}(\varepsilon, \delta = 0) \geq \Omega\left(\frac{\log W}{\varepsilon} + \frac{\log W}{\varepsilon\sqrt{m\beta}} + \frac{\log W}{m\beta}\right)$.*

*Proof.* Let $\mathcal{D}_i' = (\mathcal{D}_i)^m$ for all $i \in [W]$; note that $d_{\mathrm{KL}}(\mathcal{D}_i' \parallel \mathcal{D}_j') \leq m\beta$. By Pinsker's inequality (Lemma 23(i)), $d_{\mathrm{tv}}(\mathcal{D}_i' \parallel \mathcal{D}_j') \leq O(\sqrt{m\beta})$; furthermore, we also have the trivial bound $d_{\mathrm{tv}}(\mathcal{D}_i' \parallel \mathcal{D}_j') \leq 1$. Finally, define a task $\mathfrak{T}'$ by $\Psi_{\mathfrak{T}'}(\mathcal{D}_i') := \Psi_{\mathfrak{T}}(\mathcal{D}_i)$ for all $i \in [W]$. Now, applying Theorem 38,

$$n_1^{\mathfrak{T}'}(\varepsilon, \delta = 0) \geq \Omega\left(\frac{\log W}{\min\{1, \sqrt{m\beta}\} \cdot \varepsilon} + \frac{\log W}{m\beta}\right) = \Omega\left(\frac{\log W}{\varepsilon} + \frac{\log W}{\varepsilon\sqrt{m\beta}} + \frac{\log W}{m\beta}\right).$$

Finally, observing that $n_m^{\mathfrak{T}}(\varepsilon, \delta) = n_1^{\mathfrak{T}'}(\varepsilon, \delta)$ yields our final bound. $\qquad\square$

## E.3 Applications

Our user-level DP EM with pairwise score given above has a wide variety of applications. We now give a few examples.

### E.3.1 Agnostic PAC Learning

We start with *agnostic PAC learning*. The setting is exactly the same as in PAC learning (described in the introduction; see Theorem 4) except that we do not assume that $\mathcal{D}$ is realizable by some concept in $\mathcal{C}$. Instead, the task $\mathrm{agn}(\mathcal{C}; \alpha)$ seeks an output $c : \mathcal{X} \to \{0, 1\}$ such that $\mathrm{err}_{\mathcal{D}}(c) \leq \min_{c' \in \mathcal{C}} \mathrm{err}_{\mathcal{D}}(c') + \alpha$.

**Theorem 40.** *Let $\mathcal{C}$ be any concept class with probabilistic representation dimension $d$ (i.e., $\mathrm{PRDim}(\mathcal{C}) = d$) and let $d_{\mathrm{VC}}$ be its VC dimension. Then, for any sufficiently small $\alpha, \varepsilon > 0$ and for all $m \in \mathbb{N}$,*

$$n_m^{\mathrm{agn}(\mathcal{C};\alpha)}(\varepsilon, \delta = 0) \leq \widetilde{O}\left(\frac{d}{\varepsilon} + \frac{d}{\alpha\varepsilon\sqrt{m}} + \frac{d}{\alpha^2 m}\right),$$

---

[13]The particular version we use is a slight restatement of [ASZ21, Corollary 4].

*and*

$$n_m^{\mathrm{agn}(\mathcal{C};\alpha)}(\varepsilon, \delta = 0) \geq \tilde{\Omega}\left(\frac{d}{\varepsilon} + \frac{d_{\mathrm{VC}}}{\alpha\varepsilon\sqrt{m}} + \frac{d_{\mathrm{VC}}}{\alpha^2 m}\right).$$

Observe that our bounds are (up to polylogarithmic factors) the same except for the $d$-vs-$d_{\mathrm{VC}}$ in the last two terms. We remark that the ratio $d/d_{\mathrm{VC}}$ is not always bounded; e.g., for thresholds, $d_{\mathrm{VC}} = 2$ while $d = \Theta(\log|\mathcal{Z}|)$. A more careful argument can show that $d$ in the last two terms in the upper bound can be replaced by the (maximum) VC dimension of the probabilistic representation (which is potentially smaller); this actually closes the gap in the particular case of thresholds. However, in the general case, the VC dimension of $\mathcal{C}$ and the VC dimension of its probabilistic represetation are not necessarily equal. It remains an interesting open question to close this gap.

We now prove Theorem 40. The algorithm is similar to that in the proof of Theorem 4, except that we use the pairwise scoring function to compare the errors of the two hypotheses.

*Proof of Theorem 40.* **Algorithm.** For any two hypotheses $c, c' : \mathcal{X} \to \{0, 1\}$, let the comparison function between two concepts be $\psi_{c,c'}((x, y)) = \mathbf{1}[c(x) = y] - \mathbf{1}[c'(x) = y]$.

Let $\mathcal{P}$ denote a $(0.01\alpha, 1/4)$-PR of $\mathcal{C}$; by Lemma 21, there exists such $\mathcal{P}$ with $\mathrm{size}(\mathcal{P}) \leq \tilde{O}(d \cdot \log(1/\alpha))$. Our algorithm works as follows: Sample $\mathcal{H} \sim \mathcal{P}$ and then run the $\varepsilon$-DP pairwise scoring EM (Theorem 37) on candidate set $\mathcal{H}$ with the comparison function $\psi_{c,c'}$ defined above. The privacy guarantee follows from that of Theorem 37.

As for the utility, first observe that

$$\mathrm{pscr}_{c,c'}^{\psi}(\mathcal{D}) = \mathrm{err}_{\mathcal{D}}(c) - \mathrm{err}_{\mathcal{D}}(c'). \tag{14}$$

This means that, if we let $h^* \in \mathrm{argmin}_{h \in \mathcal{H}} \mathrm{err}_{\mathcal{D}}(h)$, then $\mathrm{scr}_{h^*}^{\psi}(\mathcal{D}) \leq 0$. Thus, Theorem 37 ensures that, w.p. 0.99, if $n \geq \tilde{\Theta}\left(\frac{\mathrm{size}(\mathcal{P})}{\varepsilon} + \frac{\mathrm{size}(\mathcal{P})}{\alpha\varepsilon\sqrt{m}} + \frac{\mathrm{size}(\mathcal{P})}{\alpha^2 m}\right) = \tilde{\Theta}\left(\frac{d}{\varepsilon} + \frac{d}{\alpha\varepsilon\sqrt{m}} + \frac{d}{\alpha^2 m}\right)$, then the output $h$ satisfies $\mathrm{scr}_h^{\psi}(\mathcal{D}) \leq 0.5\alpha$. By (14), this also means that $\mathrm{err}_{\mathcal{D}}(h) \leq 0.5\alpha + \mathrm{err}_{\mathcal{D}}(h^*)$. Finally, by the guarantee of $\mathcal{P}$, we have that $\mathrm{err}_{\mathcal{D}}(h^*) \leq 0.01\alpha + \min_{c' \in \mathcal{C}} \mathrm{err}_{\mathcal{D}}(c')$ with probability 3/4. By a union bound, we can conclude that w.p. more than 2/3, we have $\mathrm{err}_{\mathcal{D}}(h) < \alpha + \min_{c' \in \mathcal{C}} \mathrm{err}_{\mathcal{D}}(c')$.

**Lower Bound.** The lower bound $\Omega(d/\varepsilon)$ was shown in [GKM21] (and holds even for realizable PAC learning). To derive the lower bound $\Omega\left(\frac{d_{\mathrm{VC}}}{\alpha\varepsilon\sqrt{m}} + \frac{d_{\mathrm{VC}}}{\alpha^2 m}\right)$, we will use DP Fano's inequality (Lemma 39). To do so, recall that $\mathcal{C}$ having VC dimension $d_{\mathrm{VC}}$ means that there exist $\{x_1, \ldots, x_{d_{\mathrm{VC}}}\}$ that is shattered by $\mathcal{Z}$. From Theorem 24, there exist $u_1, \ldots, u_W \in \{0, 1\}^{d_{\mathrm{VC}}}$ where $W = 2^{\Omega(d_{\mathrm{VC}})}$ such that $\|u_i - u_j\|_0 \geq \kappa \cdot d_{\mathrm{VC}}$ for some constant $\kappa$. For any sufficiently small $\alpha < \kappa/8$, we can define the distribution $\mathcal{D}_i$ for $i \in [W]$ by

$$\mathcal{D}_i((x_\ell, y)) = \frac{1}{2d_{\mathrm{VC}}} \cdot \left(1 + \frac{4\alpha}{\kappa} \cdot (2\mathbf{1}[(u_i)_\ell = y] - 1)\right),$$

for all $\ell \in [W]$ and $y \in \{0, 1\}$. Next, notice that, for any $i \in [W]$, we may select $c_i$ such that $\mathrm{err}_{\mathcal{D}}(c_i) = \frac{1}{2}\left(1 - \frac{4\alpha}{\kappa}\right)$.

Furthermore, for any $i \neq j \in [W]$ and any hypothesis $h : \mathcal{X} \to \{0, 1\}$, we have

$$\mathrm{err}_{\mathcal{D}_i}(h) + \mathrm{err}_{\mathcal{D}_j}(h) \geq \left(1 - \frac{4\alpha}{\kappa}\right) + \frac{4\alpha}{\kappa} \cdot \frac{\kappa \cdot d_{\mathrm{VC}}}{d_{\mathrm{VC}}} = \left(1 - \frac{4\alpha}{\kappa}\right) + 4\alpha.$$

This means that $\mathrm{err}_{\mathcal{D}_i}(h) > \frac{1}{2}\left(1 - \frac{4\alpha}{\kappa}\right) + 2\alpha$ or $\mathrm{err}_{\mathcal{D}_j}(h) > \frac{1}{2}\left(1 - \frac{4\alpha}{\kappa}\right) + 2\alpha$. In other words, we have $\Psi_{\mathrm{agn}(\mathcal{C};\alpha)}(\mathcal{D}_i) \cap \Psi_{\mathrm{agn}(\mathcal{C};\alpha)}(\mathcal{D}_j) = \emptyset$. Finally, we have

$$d_{\mathrm{KL}}(\mathcal{D}_i \| \mathcal{D}_j) \leq d_{\chi^2}(\mathcal{D}_i \| \mathcal{D}_j)$$
$$= \sum_{\ell \in [d_{\mathrm{VC}}], y \in \{0,1\}} \frac{(\mathcal{D}_i((x, y)) - \mathcal{D}_j((x, y)))^2}{\mathcal{D}_j((x, y))}$$
$$\leq \sum_{\ell \in [d_{\mathrm{VC}}], y \in \{0,1\}} \frac{O(\alpha/d_{\mathrm{VC}})^2}{1/4d_{\mathrm{VC}}}$$

$$= \sum_{\ell \in [d_{\text{VC}}]} O(\alpha^2/d_{\text{VC}})$$
$$\leq O(\alpha^2).$$

Plugging this into Lemma 39, we get that $n_m^{\text{agn}(\mathcal{C};\alpha)} \geq \Omega \left( \frac{d_{\text{VC}}}{\varepsilon \alpha \sqrt{m}} + \frac{d_{\text{VC}}}{m\alpha^2} \right)$ as desired. $\qquad \square$

### E.3.2 Private Hypothesis Selection

In *Hypothesis Selection problem*, we are given a set $\mathfrak{P}$ of hypotheses, where each $\mathcal{P} \in \mathfrak{P}$ is a distribution over some domain $\mathcal{Z}$. The goal is, for the underlying distribution $\mathcal{D}$, to output $\mathcal{P}^* \in \mathfrak{P}$ that is the closest (in TV distance) to $\mathcal{D}$. We state below the guarantee one can get from our approach:

**Theorem 41.** *Let $\alpha, \beta \in (0, 0.1)$. Then, for any $\varepsilon > 0$, there exists an $\varepsilon$-user-level DP algorithm for Hypothesis Selection that, when each user has $m$ samples and $\min_{\mathcal{P}' \in \mathfrak{P}} d_{\text{tv}}(\mathcal{P}', \mathcal{D}) \leq 0.1\alpha$, with probability $1 - \beta$, outputs $\mathcal{P} \in \mathfrak{P}$ such that $d_{\text{tv}}(\mathcal{P}, \mathcal{D}) \leq \alpha$ as long as*

$$n \geq \tilde{\Theta} \left( \frac{\log(|\mathfrak{P}|/\beta)}{\varepsilon} + \frac{\log(|\mathfrak{P}|/\beta)}{\alpha\varepsilon\sqrt{m}} + \frac{\log(|\mathfrak{P}|/\beta)}{\alpha^2 m} \right).$$

For $m = 1$, the above theorem essentially match the item-level DP sample complexity bound from [BKSW19]. The proof also proceeds similarly as theirs, except that we use the user-level DP EM rather than the item-level DP one.

*Proof Theorem 41.* We use the so-called *Scheffé score* similar to [BKSW19]: For every $\mathcal{P}, \mathcal{P}' \in \mathfrak{P}$, we define $\mathcal{W}_{\mathcal{P}, \mathcal{P}'}$ as the set $\{z \in \mathcal{Z} \mid \mathcal{P}(z) > \mathcal{P}'(z)\}$ and let

$$\psi_{\mathcal{P}, \mathcal{P}'}(z) = \mathcal{P}(\mathcal{W}_{\mathcal{P}, \mathcal{P}'}) - \mathbf{1}[z \in \mathcal{W}_{\mathcal{P}, \mathcal{P}'}].$$

We then use the $\varepsilon$-user-level DP pairwise scoring EM (Theorem 37). The privacy guarantee follows immediately from the theorem.

For the utility analysis, let $\mathcal{P}^* \in \text{argmin}_{\mathcal{P} \in \mathfrak{P}} d_{\text{tv}}(\mathcal{P}, \mathcal{D})$. Recall from our assumption that $d_{\text{tv}}(\mathcal{P}^*, \mathcal{D}) \leq 0.1\alpha$. Furthermore, observe that

$$\text{pscr}_{\mathcal{P}^*, \mathcal{P}}^{\psi}(\mathcal{D}) = \mathcal{P}^*(\mathcal{W}_{\mathcal{P}^*, \mathcal{P}}) - \mathcal{D}(\mathcal{W}_{\mathcal{P}^*, \mathcal{P}}) \leq d_{\text{tv}}(\mathcal{D}, \mathcal{P}^*) \leq 0.1\alpha.$$

Moreover, for every $\mathcal{P} \in \mathfrak{P}$, we have

$$\begin{aligned}
\text{pscr}_{\mathcal{P}, \mathcal{P}^*}^{\psi}(\mathcal{D}) &= \mathcal{P}(\mathcal{W}_{\mathcal{P}, \mathcal{P}^*}) - \mathcal{D}(\mathcal{W}_{\mathcal{P}, \mathcal{P}^*}) \\
&= (\mathcal{P}(\mathcal{W}_{\mathcal{P}, \mathcal{P}^*}) - \mathcal{P}^*(\mathcal{W}_{\mathcal{P}, \mathcal{P}^*})) - (\mathcal{D}(\mathcal{W}_{\mathcal{P}, \mathcal{P}^*}) - \mathcal{P}^*(\mathcal{W}_{\mathcal{P}, \mathcal{P}^*})) \\
&= d_{\text{tv}}(\mathcal{P}, \mathcal{P}^*) - (\mathcal{D}(\mathcal{W}_{\mathcal{P}, \mathcal{P}^*}) - \mathcal{P}^*(\mathcal{W}_{\mathcal{P}, \mathcal{P}^*})) \\
&\geq d_{\text{tv}}(\mathcal{P}, \mathcal{P}^*) - d_{\text{tv}}(\mathcal{P}^*, \mathcal{D}) \\
&\geq d_{\text{tv}}(\mathcal{P}, \mathcal{P}^*) - 0.1\alpha.
\end{aligned}$$

Thus, applying the utility guarantee from Theorem 37, we can conclude that, with probability $1 - \beta$, the algorithm outputs $\mathcal{P}$ such that $d_{\text{tv}}(\mathcal{P}, \mathcal{D}) \leq 0.5\alpha + 0.1\alpha < \alpha$ as desired. $\qquad \square$

### E.3.3 Distribution Learning

Private hypothesis selection has a number of applications, arguably the most prominent one being *distribution learning*. In distribution learning, there is a family $\mathfrak{P}$ of distributions. The underlying distribution $\mathcal{D}$ comes from this family. The task $\text{dlearn}(\mathfrak{P}; \alpha)$ is to output a distribution $\mathcal{Q}$ such that $d_{\text{tv}}(\mathcal{Q}, \mathcal{D}) \leq \alpha$, where $\alpha > 0$ denotes the accuracy parameter.

For convenient, let us defined *packing* and *covering* of a family of distributions:

- A family $\mathfrak{Q}$ of distributions is an $\alpha$-*cover* of a family $\mathfrak{P}$ of distributions under distance measure $d$ iff, for all $\mathcal{P} \in \mathfrak{P}$, there exists $\mathcal{Q} \in \mathfrak{Q}$ such that $d(\mathcal{Q}, \mathcal{P}) \leq \alpha$.
- A family $\mathfrak{Q}$ of distributions is an $\alpha$-*packing* of a family $\mathfrak{P}$ of distributions under distance measure $d$ iff, $\mathfrak{Q} \subseteq \mathfrak{P}$ and for all distinct $\mathcal{Q}, \mathcal{Q}' \in \mathfrak{Q}$, we have $d(\mathcal{Q}, \mathcal{Q}') \geq \alpha$.

The size of the cover / packing is defined as $\mathrm{size}(\mathfrak{Q}) := \log(|\mathfrak{Q}|)$. The diameter of $\mathfrak{Q}$ under distance $d$ is defined as $\max_{\mathcal{Q},\mathcal{Q}'\in\mathfrak{Q}} d(\mathcal{Q},\mathcal{Q}')$.

The following convenient lemma directly follows from Theorem 41 and Lemma 39.

**Theorem 42** (Distribution Learning—Arbitrary Family). *Let $\mathfrak{P}$ be any familiy of distributions, and let $\alpha, \beta, \varepsilon > 0$ be sufficiently small. If there exists an $0.1\alpha$-cover of $\mathfrak{P}$ under the TV distance of size $L$, then*

$$n_m^{\mathrm{dlearn}(\mathfrak{P};\alpha)}(\varepsilon, \delta = 0) \leq \widetilde{O}\left(\frac{L}{\varepsilon} + \frac{L}{\alpha\varepsilon\sqrt{m}} + \frac{L}{\alpha^2 m}\right).$$

*Furthermore, if there exists a family $\mathfrak{Q}$ of size $L$ that is an $3\alpha$-packing of $\mathfrak{P}$ under the TV distance and has diameter at most $\beta$ under the KL-divergence, then*

$$n_m^{\mathrm{dlearn}(\mathfrak{P};\alpha)}(\varepsilon, \delta = 0) \geq \tilde{\Omega}\left(\frac{L}{\varepsilon} + \frac{L}{\varepsilon\sqrt{m\beta}} + \frac{L}{m\beta}\right).$$

In theorems below, we abuse the notations $\tilde{\Theta}, \widetilde{O}$ and also use them to suppress terms that are polylogarithmic in $1/\alpha, d, R$, and $\kappa$.

**Discrete Distributions.** First is for the task of discrete distribution learning on domain $[k]$ (denoted by $\mathrm{DD}k(\alpha)$), which has been studied before in [LSY$^+$20]. In this case, $\mathfrak{P}$ consists of all distributions over the domain $[k]$. We have the following theorem:

**Theorem 43** (Distribution Learning—Discrete Distributions). *For any sufficiently small $\alpha, \varepsilon > 0$ and for all $m \in \mathbb{N}$, we have*

$$n_m^{\mathrm{DD}k(\alpha)}(\varepsilon, \delta = 0) = \tilde{\Theta}\left(\frac{k}{\varepsilon} + \frac{k}{\alpha\varepsilon\sqrt{m}} + \frac{k}{\alpha^2 m}\right).$$

*Proof.* **Upper bound.** It is simple to see that there exists an $0.1\alpha$-cover of the family of size $O(k \cdot \log(k/\alpha))$: We simply let the cover contain all distributions whose probability mass at each point is a multiple of $\lfloor 10k/\alpha \rfloor$. Plugging this into Theorem 42 yields the desired upper bound.

**Lower bound.** As for the lower bound, we may use a construction similar to that of [ASZ21] (and also similar to that in the proof of Theorem 40). Assume w.l.o.g. that $k$ is even. From Theorem 24, there exist $u_1, \ldots, u_W \in \{0,1\}^{k/2}$ where $W = 2^{\Omega(k)}$ such that $\|u_i - u_j\|_0 \geq \kappa \cdot k$ for some constant $\kappa$. For any sufficiently small $\alpha < \kappa/16$, we can define the distribution $\mathcal{D}_i$ for $i \in [W]$ by

$$\mathcal{D}_i(2\ell - 1) = \frac{1}{2k}\left(1 + \frac{8\alpha}{\kappa} \cdot \mathbf{1}[(u_i)_\ell = 0]\right)$$

$$\mathcal{D}_i(2\ell) = \frac{1}{2k}\left(1 - \frac{8\alpha}{\kappa} \cdot \mathbf{1}[(u_i)_\ell = 0]\right),$$

for all $\ell \in [k/2]$. By a similar calculation as in the proof of Theorem 40, we have that it is an $(3\alpha)$-packing under TV distance and its diameter under KL-divergence is at most $O(\alpha^2)$. Plugging this into Theorem 42 then gives the lower bound. $\square$

Interestingly, the user complexity matches those achieved via *approximate-DP* algorithms from [LSY$^+$20, NME22], except for the first term (i.e., $k/\varepsilon$ whereas the best known $(\varepsilon, \delta)$-DP algorithm of [NME22] works even with $\log(1/\delta)/\varepsilon$ instead. In other words, when $m$ is intermediate, the user complexity between the pure-DP and approximate-DP cases are the same. Only for large $m$ that approximate-DP helps.

**Product Distributions.** In this case, $\mathfrak{P}$ is a product distribution over the domain $[k]^d$ (denoted by $\mathrm{PD}(k, d; \alpha)$). We have the following theorem:

**Theorem 44** (Distribution Learning—Product Distributions). *For any sufficiently small $\alpha, \varepsilon > 0$ and for all $m \in \mathbb{N}$, we have*

$$n_m^{\mathrm{PD}(k,d;\alpha)}(\varepsilon, \delta = 0) = \tilde{\Theta}\left(\frac{kd}{\varepsilon} + \frac{kd}{\alpha\varepsilon\sqrt{m}} + \frac{kd}{\alpha^2 m}\right).$$

*Proof.* **Upper Bound.** Similar to before, we simply take $\mathfrak{Q}$ to be a $0.1\alpha$-cover (under TV distance) of the product distributions, which is known to have size at most $O(kd \cdot \log(kd/\alpha))$ [BKSW19, Lemma 6.2].

**Lower Bound.** Acharya et al. [ASZ21, Proof of Theorem 11] showed that there exists a family $\mathfrak{Q}$ of product distributions over $[k]^d$ that is a $(3\alpha)$-packing under TV distance and its diameter under KL-divergence is at most $O(\alpha^2)$ of size $\Omega(kd)$. Theorem 42 then gives the lower bound. $\qquad\square$

**Gaussian Distributions: Known Covariance.** Next, we consider Gaussian distributions $\mathcal{N}(\mu, \Sigma)$, where $\mu \in \mathbb{R}^d, \Sigma \in \mathbb{R}^{d \times d}$, under the assumption that $\|\mu\| \leq R$ and $\Sigma = I$[14]. Let $\mathrm{Gauss}(R, d; \alpha)$ denote this task, where $\alpha$ is again the (TV) accuracy. For this problem, we have:

**Theorem 45** (Distribution Learning—Gaussian Distributions with Known Covariance). *For any sufficiently small $\alpha, \varepsilon > 0$ and for all $m \in \mathbb{N}$, we have*

$$n_m^{\mathrm{Gauss}(R,d;\alpha)}(\varepsilon, \delta = 0) = \tilde{\Theta}\left(\frac{d}{\varepsilon} + \frac{d}{\alpha\varepsilon\sqrt{m}} + \frac{d}{\alpha^2 m}\right).$$

*Proof.* **Upper Bound.** It is known that this family of distribution admits a $(0.1\alpha)$-cover (under TV distance) of size $O(d \cdot \log(dR/\alpha))$ [BKSW19, Lemma 6.7]. This, together with Theorem 42, gives the upper bound.

**Lower Bound.** Again, Acharya et al. [ASZ21, Proof of Theorem 12] showed that there exists a family $\mathfrak{Q}$ of isotropic Gaussian distributions with $\|\mu\| \leq O(\sqrt{\log(1/\alpha)})$ that is a $(3\alpha)$-packing under TV distance and its diameter under KL-divergence is at most $O(\alpha^2)$ of size $\Omega(d)$. Theorem 42 then gives the lower bound. $\qquad\square$

We note that this problem has already been (implicitly) studied under user-level approximate-DP in [NME22],[15] who showed that $n_m^{\mathrm{Gauss}(R,d;\alpha)}(\varepsilon, \delta > 0) \leq \tilde{O}\left(\frac{d}{\alpha\varepsilon\sqrt{m}} + \frac{d}{\alpha^2 m}\right)$. Again, it is perhaps surprising that our pure-DP bound nearly matches this result, except that we have the extra first term (i.e., $d/\varepsilon$).

**Gaussian Distributions: Unknown Bounded Covariance.** Finally, we consider the case where the covariance is also unknown but is assumed to satisfied $I \preceq \Sigma \preceq \kappa I$. We use $\mathrm{Gauss}(R, d, \kappa; \alpha)$ to denote this task.

**Theorem 46** (Distribution Learning—Gaussian Distributions with Unknown Bounded Covariance). *Let $\kappa > 1$ be a constant. For any sufficiently small $\alpha, \varepsilon > 0$ and for all $m \in \mathbb{N}$, we have*

$$n_m^{\mathrm{Gauss}(R,d,\kappa;\alpha)}(\varepsilon, \delta = 0) = \tilde{\Theta}_\kappa\left(\frac{d^2}{\varepsilon} + \frac{d^2}{\alpha\varepsilon\sqrt{m}} + \frac{d^2}{\alpha^2 m}\right).$$

*Proof.* **Upper Bound.** This follows from Theorem 42 and a known upper bound of $O(d^2 \log(d\kappa/\alpha) + d \cdot \log(dR/\alpha))$ on the size of a $(0.1\alpha)$-cover (under TV distance) of the family [BKSW19, Lemma 6.8].

**Lower Bound.** Devroye et al. [DMR20] showed[16] that, for $\kappa \geq 1 + O(\alpha)$, there exists a $3\alpha$-packing under TV distance whose diameter under the KL-divergence is at most $O(\alpha^2)$ of size $\Omega(d^2)$. This, together with Theorem 42, gives the lower bound. $\qquad\square$

We remark that it is simple to see that Gaussian with unbounded mean or unbounded covariance cannot be learned with pure (even user-level) DP using a finite number of examples.

---

[14] Or equivalently that $\Sigma$ is known; since such an assumption is sufficient to rotate the example to isotropic positions.

[15] Actually, Narayanan et al. [NME22] studied the mean estimation problem, but it is not hard to see that an algorithm for mean estimation also provides an algorithm for learning Gaussian distributions.

[16] Specifically, this is shown in [DMR20, Proposition 3.1]; their notation "$m$" denote the number of non-zero (non-diagonal) entries of the covariance matrix which can be set to $\Omega(d^2)$ for our purpose.

