# OpenReview forum: "User-Level Differential Privacy With Few Examples Per User"
_NeurIPS.cc/2023/Conference — NeurIPS 2023 oral_

### Official Review · Reviewer_iTdY · 2023-07-06

**Soundness:** 3 good
**Presentation:** 4 excellent
**Contribution:** 4 excellent
**Rating:** 7
**Confidence:** 3

**Summary:**

This work designs a generic algorithm that transforms an item-level DP algorithm into a user-level DP one with $\sqrt{m}$ improvement in user complexity. It recovers previous user complexity bounds on various learning tasks, and the transformation works in the example sparse setting, i.e. does not require sufficient number of samples per user.

**Strengths:**

1. Compared to Ghazi et.al. 2021 which requires sufficient number of samples per user, the generic transformation from item-level to user-level DP in this work applies to the example scarce setting. The authors derive new bounds in the example-scarce setting for PAC learning.
2. The authors recover the $\sqrt{m}$ factor in many known bounds for various learning tasks through a generic algorithm.
3. The writing is clear and easy to follow.

**Weaknesses:**

1. The proposed algorithm is not computationally efficient.
2. Although $\sqrt{m}$ matches the tight bounds for a variety of problems, it seems that the lower bound is not addressed in this work.
3. In Theorem 16, it might be helpful to be specific about how sufficiently small should $\varepsilon', \delta'$ be.

Minor issues:
1. In Lemma 4, I feel that it should be $A\simeq C$ instead of $B$ at the end of the line.
2. In line 242, it appears that the $\delta'$ inside the big O should be $\delta''$, and there should be a factor of $(1+e^{\varepsilon''})$ for the $\delta''$ term according to Lemma 4 (even though it does not affect the final bound on $\delta'$).


**Questions:**

1.  I wonder if it is possible show a $\Omega(\sqrt{m})$ lower bound as well under some non-trivial conditions?
2.  In Theorem 16, how sufficiently small are $\varepsilon', \delta'$? Is it for some fixed small constant, or some expression that depends on other parameters?

**Limitations:**

Limitations and potential impacts are adequately addressed.

---

> ### Author Rebuttal · Authors · 2023-08-08
>
> Thank you for your review and questions. Thank you also for pointing out the typos. We will fix them in the revision. Please find our answers below.
>
> ## Re $\\sqrt{m}$ lower bound:
>
>
> Interestingly, such a lower bound does not necessarily hold for all problems, even natural ones. For example, taking the class of all functions on $\\mathcal{X} = [d]$, our pure-DP upper bound (Theorem 3) is in fact better than our approximate-DP upper bound (Corollary 2) for the following regime of parameters: $\\epsilon=1,\\alpha=1/m, m \\leq d$, which gives $O(d)$ vs $\\tilde{O}(d\\sqrt{m})$. Thus, the $\\sqrt{m}$ lower bound does not hold in this setting. We will add more discussion on this in the revision.
>
> ## Re Privacy Parameters in Theorem 16:
>
>
> The $\\epsilon’, \\delta’, \\epsilon’ r \\sqrt{m \\log(n/\\beta\\delta’)}$ values here only have to be smaller than some absolute constant independent of the other parameters.

---

### Official Review · Reviewer_KRgS · 2023-07-06

**Soundness:** 4 excellent
**Presentation:** 3 good
**Contribution:** 4 excellent
**Rating:** 8
**Confidence:** 4

**Summary:**

This paper studies the problem of user-level differential privacy, giving a generic conversion for any item-level approx DP algorithm to a user-level approx DP algorithm, and a clipped scoring function that allows application of the exponential mechanism to obtain user-level privacy. All results are applicable in the few-samples-per-user setting, significantly extending results beyond the case in which users have enough samples to solve a task independently.

**Strengths:**

This work significantly improves our understanding of when user-level privacy is achievable, though it is unfortunate the transformation from item-level to user-level privacy isn't computationally efficient. The intermediate result that approx DP => sample perfect generalization is also a meaningful contribution to our understanding of the relationships between useful algorithmic stability notions.



**Weaknesses:**

In terms of substance, I think this work is very strong, but the clarity of presentation could maybe be improved a little. That said, I do think the authors did a good job overall of assisting the reader with interpretation of their results.

Notes:

The first paragraph of the technical overview for the approximate DP result was a bit confusing.

Page 4: “it is sufficient get new user-level pure-DP”

Page 6: 	“For example, g is not Lipschitz everywhere anyway anymore”




**Questions:**

There's a related result from BGHILPSS23 showing that approx DP implies perfect generalization, following from the connections approx DP => one-way perfect generalization, one-way perfect generalization => replicability, replicability => perfect generalization. However, my understanding is that their result is weaker than the one given in this work because it 1) requires correlated sampling and is therefore not necessarily computationally efficient and 2) the transformation from replicability to perfect generalization changes the output distribution of the DP algorithm, whereas the result in this work proves perfect generalization for the same approx DP algorithm A. Is this a correct comparison of these results, or is there any notable distinction between the resulting PG parameters obtained from both results?

It would be very interesting to understand whether the computational inefficiency is in fact necessary.

**Limitations:**

Yes, the authors adequately address societal impact of their work.

---

> ### Author Rebuttal · Authors · 2023-08-08
>
> Thank you for your review and questions. Please find our answers below.
>
> ## Re [BBH+23]:
> The main distinction between the two results is that we show that if an algorithm satisfies DP, then that very same algorithm also satisfies PG. On the other hand, [BBH+23] shows that if an algorithm satisfies DP, then there is *a different algorithm* that satisfies PG and maintains a similar utility guarantee.
>
>
> The parameters of the two approaches are different. In [BBH+23], the reduction loses a roughly square factor in the sample complexity. That is, if we start with a DP algorithm with sample complexity $n$, then applying the reduction from [BBH+23] will result in a PG algorithm with sample complexity $\\approx n^2$ (for the same utility guarantee). On the other hand, we do not incur such a loss, since we prove the property directly on the original algorithm.
>
> ## Re computational inefficiency:
>
> We agree that this is a great question and we had included it in our conclusions as well (lines  347-353). We feel this is necessary for such a generic reduction; however, we do not have a formal proof of such a statement, as it is quite challenging (in general) to prove computational lower bounds for DP algorithms.

---

> > ### Author Response · Authors · 2023-08-15
> >
> > Note: Please disregard the "the parameters of two approaches ..." paragraph in our rebuttal above. As pointed out by reviewer pUZs, the two approaches actually give similar parameters. We stress that the first paragraph ("The main distinction ...") still holds.

---

> > ### Comment · Reviewer_KRgS · 2023-08-18
> >
> > Thank you for the response!

---

### Official Review · Reviewer_pUZs · 2023-07-06

**Soundness:** 4 excellent
**Presentation:** 3 good
**Contribution:** 3 good
**Rating:** 7
**Confidence:** 4

**Summary:**

This paper establishes new generic conversions from item-level DP algorithms to user-level DP algorithms, continuing a line of work initiated by Ghazi, Kumar, and Manurangsi. Previous state-of-the-art due to BGHILPSS '23 shows that any item-level $(\varepsilon, \delta)$-DP algorithm using N samples can be converted to a user-level DP algorithm with $n = O(\log(1/\delta)/\varepsilon$ users and $m \approx N^2$ samples per user. The present submission observes that this previous result is just one end of a spectrum of possible item-to-user-level conversions, and generalizes it to hold for the full range of $1 \le m \le N^2$. That is, the main result (somewhat simplifying) shows a conversion from any item-level DP algorithm to a user-level DP algorithm with $n \approx (N/\sqrt{m}) \cdot \log^{O(1)} ({1/\delta})/\varepsilon^2$ users and $m$ samples per user. In addition to generalizing the conversions of GKM'21 and BGHILPSS'23 to the regime of few examples per user, this result unifies previous work on specific problems such as mean estimation, stcochastic convex optimization, and discrete distribution learning for which a $n \propto 1/\sqrt{m}$ dependence was previously observed.

As an auxiliary result, the paper also gives a tight characterization of the sample complexity of user-level pure $(\varepsilon, 0)$-differentially private PAC learning in terms of the probabilistic representation dimension.

**Strengths:**

- This is a nice contribution to the emerging general theory around user-level DP learning, in particular, building a bridge between one thread of work on general item-level vs. user-level connections and another on understanding the sample complexity of user-level DP for specific problems.

- As part of their analysis, the authors more carefully investigate the connection between DP and perfect generalization, in particular giving a clean solution to an old open question of CNLRW'16 that is likely to have further applications.

**Weaknesses:**

- The main technical ideas going into the algorithms and their analysis appeared in prior work on this topic. (E.g., the passage from DP to perfect generalization appears in GKM'21/BGHILPSS'23, and a similar PTR-based argument appeared in KL'21/GKKMMZ'23). This paper's technical contribution is to carefully refine each of these steps and piece them together in the right way.

- Neither of the paper's new algorithms is computationally efficient in general, even if the task admits a computationally efficient item-level DP learner. Note that this is the case for previous general item-to-user-level conversions as well.

- The final main result (Theorem 9) is somewhat messy to state and potentially suboptimal in its dependence on $\log(1/\delta)$ and $\varepsilon$. This is likely in part due to stacking various generic tools (DP => PG, amplification by subsampling, PTR) that could potentially be avoided with a simpler algorithm.

**Questions:**

- The prior work of BBGHLISS'23 also claims to have resolved the open question of CLNRW'16 on DP vs. perfect generalization, but I believe they did so in a weaker sense. Namely, they only showed that every DP algorithm admits a PG algorithm solving the same problem; whereas this submission shows that the original DP algorithm itself perfectly generalizes. This is an important distinction that I think is worth highlighting.

- I wasn't really able to match the informal description of the proof of Lemma 13 (lines 214-226) to the formal proof in appendix B. In particular, the new contribution of the formal proof seems to be to apply a nice symmetrization argument to the previous results of RRST'16/BGHILPSS'23 that held for a one-sided version of the DP -> PG connection. Does the informal argument in the main body of the text reflect what's going on under the hood of these prior results? And if so, does it suggest a simpler / more direct argument that doesn't go through max-information etc.?

- I'd suggest renaming "local deletion DP" (Definition 15) to something else due to the (unfortunate) naming collision that may suggest this definition has to do with local DP. Maybe something like "pointwise deletion DP"?

**Limitations:**

Limitations (overlapping with points identified as weaknesses) are nicely described as directions for future work in the conclusion.

---

> ### Author Rebuttal · Authors · 2023-08-08
>
> Thank you for your detailed review and questions. Please find our answers below.
>
> ## Re [BBH+23]:
> Thank you for your suggestion. We will add this to the discussion. We also wish to note that their reduction loses a roughly square factor in the sample complexity. That is, if we start with a DP algorithm with sample complexity $n$, then applying the reduction from [BBH+23] will result in a perfect generalization algorithm with sample complexity $\\approx n^2$ (for the same utility guarantee). On the other hand, we do not incur such a loss, since we prove the property directly on the original algorithm.
>
> ## Description vs formal proof of Lemma 13:
> We think that the description still roughly reflects the full proof. Namely, while we do indeed use the one-sided perfect generalization result from [BBH+23], we are not aware of a proof using that directly without the aforementioned strengthened version of McDiarmid’s inequality. In particular, in our proof, we need Lemma 35 which is proved using Theorem 29; the latter is indeed the aforementioned result from [Kut02] mentioned in the outline. We will make sure to also mention the use of the one-sided result in the revision for clarity.
>
> ## Local deletion DP:
>
> Thank you for your suggestion. We will consider changing the name in the revision.

---

> > ### Comment · Reviewer_pUZs · 2023-08-14
> >
> > Thank you for the additional clarifications.
> >
> > > We also wish to note that their reduction loses a roughly square factor in the sample complexity. That is, if we start with a DP algorithm with sample complexity $n$, then applying the reduction from [BBH+23] will result in a perfect generalization algorithm with sample complexity
> >  $\approx n^2$ (for the same utility guarantee). On the other hand, we do not incur such a loss, since we prove the property directly on the original algorithm.
> >
> > Assuming you're referring to BGHILPSS'23, I believe that both papers achieve roughly the same quantitative parameters. Combining Corollary 3.18 with Theorem 3.19 from the arXiv version of their paper (https://arxiv.org/pdf/2303.12921.pdf) roughly shows that a $(1/\sqrt{m}, \delta)$-DP algorithm using $m$ samples can be converted to a $(\delta, \varepsilon, \delta)$-PG algorithm using $m/\varepsilon^2$ samples. This reflects the same $\sqrt{m}$-factor blowup in parameters that one gets by going from DP to PG as appears in your work. The quadratic sample complexity blowup described in your comment follows by taking an $(\varepsilon, \delta)$-DP using $n$ samples, applying privacy amplification by subsampling to turn it into a $(1/\sqrt{m}, \delta)$-DP algorithm using $m \approx \varepsilon^2 n^2$ samples, and then applying the above transformation to get a $(\delta, \varepsilon, \delta)$-PG algorithm using $\approx n^2$ samples. Similarly applying privacy amplification by subsampling to your Theorem 34 would also incur a quadratic sample complexity blowup when going from a $(\varepsilon, \delta)$-DP algorithm to a $(\delta, \varepsilon, \delta)$-PG algorithm.
> >
> > I'd be happy to be corrected if I've missed something or if you were referring to a different set of parameters!

---

> > > ### Author Response · Authors · 2023-08-15
> > >
> > > Thank you for pointing this out. Yes, you are absolutely correct and really sorry for the confusion caused by our rebuttal. We will make sure that this is very clear in the revision.
> > >
> > > Nevertheless, we would like to stress that the approach in [BBH+23] does not seem to be able to prove a similar statement as our main theorem (Theorem 9). The reason in that their compilation to achieve PG or DP (Algorithm 2) requires the entire algorithm to be replicable. This is not necessarily possible, especially in the regime where $m \ll n$--i.e. "few examples per user".

---

### Official Review · Reviewer_Hupm · 2023-07-20

**Soundness:** 4 excellent
**Presentation:** 4 excellent
**Contribution:** 4 excellent
**Rating:** 8
**Confidence:** 3

**Summary:**

This work develops novel generic algorithms that convert algorithms under item-level differential privacy (DP) to user-level DP in the low user sample regime, where each user holds a small number of samples, and obtains new results for user sample complexity (aka., min # users to achieve a certain utility of the algorithm). Based on a novel observation that connects sample perfect generalization to local deletion DP, the proposed algorithm under approximate-DP leads to tight user sample complexity bound for several privacy-preserving machine learning applications. The proposed algorithm under pure-DP is based on the exponential mechanism with a new score function and it also leads to tight and improved user sample complexity bounds for several applications.

**Strengths:**

The improved user sample complexity results and proposed algorithms under both approximate DP and pure DP are interesting and are widely applicable to several important privacy preserving machine learning tasks.

The paper is well presented and intriguing to read. The notations are well-defined throughout the paper. Problems and results are well defined and clearly stated. The contribution and overview of techniques are well summarized.

The paper tries hard to give the readers intuition on the techniques developed and does a good job explaining the observations.

The paper also explains well its difference from the previous works, especially why the techniques from previous works do not apply when the number of samples per user is low.


**Weaknesses:**

The only places that might need a bit more explanation is why local deletion DP is natural to consider based on sample perfect generalization results, and how local deletion DP is then converted to user level DP?

**Questions:**

This might be a stupid question. How is the randomness of the private algorithms considered in this work? Do we assume that a random seed is given?

Is line 9 in Algorithm 1 for removing the polylog dependence on $n$ users in the proof?

What are the major differences between pure DP and approximate DP in changing from item-level DP to user-level DP? What are the challenges that make the techniques to convert from item-level DP to user-level DP for approximate DP not applicable to pure DP?

In which applications/problems is the user sample complexity bound from this work not tight?

Minor issues: $x_{1, n}$ in line 31 $\Rightarrow$ $x_{1, m}$.

$\mathbf{x}_{T}$ in line 265

$\Rightarrow$ $\mathbf{x}_{-T}$.

Might be clearer to add “for some chosen constant $\kappa$” to the end of line 272.

**Limitations:**

Limitations (esp. the runtime of the proposed algorithm) are well discussed in the paper.

---

> ### Author Rebuttal · Authors · 2023-08-08
>
> Thank you for your detailed review and questions. Please find our answers below.
>
> ## Re randomness:
> We are in the standard setting in DP where we assume that the algorithm can sample fresh random coins and that these random coins are not accessible by the adversary.
>
> ## Line 9 Algorithm 1:
> This line of the algorithm is not required if we want just the privacy guarantee. However, we need it due to a technicality in our utility analysis: we need to make sure that in the “good” event, we run $\mathbb{A}$ on a random subset of the input samples so that the distribution of the output is (almost) the same as running $\mathbb{A}$ on i.i.d. samples from $\mathcal{D}$.  (See the proof of Lemma 18 for more details.)
>
> ## Pure-vs-Approximate DP:
> The main challenge in using the approximate-DP approach for pure-DP is that the propose-test-release technique does not work for pure-DP. Namely, even if we know that the local (deletion) sensitivity of a function is very small up to a very large distance, it is still not possible to add a small noise and achieve pure-DP. (For pure-DP, we would need either a global bound or at least a bound that is “smooth” enough.)
>
> At a higher level, the best sample complexity for pure-DP also often contains a privacy-dependent term that does not converge to zero as $m \\to \\infty$. (For example, the first term $d/\\epsilon$ in Theorem 3.) On the other hand, such a term does not appear in approximate-DP. Such a behavior suggests an inherent difference between the two settings.
>
> ## Tightness:
>
> This is a great question. We do not have a general characterization of the tightness of our results, although, as explained in the Introduction, our results are nearly tight for many problems (up to dependencies on $\\log(1/\\delta), \\epsilon$). Interestingly, however, the approximate-DP approach is *not* tight for PAC learning for some classes. For example, taking the class of all functions on $\\mathcal{X} = [d]$, our pure-DP upper bound (Theorem 3) is in fact better than our approximate-DP upper bound (Corollary 2) for some regime of parameters ($\\epsilon=1,\\alpha=1/m, m \\leq d$ gives $O(d)$ vs $\\tilde{O}(d\sqrt{m})$). We will add more discussion on this in the revision.
>
>
> Thank you for pointing out the typos; we will fix them in the revision.

---

> > ### Comment · Reviewer_Hupm · 2023-08-12
> >
> > Thank you very much for the response!

---

### Decision · Program_Chairs · 2023-09-21

**Decision:**

Accept (oral)

**Comment:**

This paper gives a generic conversion for any N-sample item-level approx DP algorithm to a user-level approx DP algorithm where each user has m samples. Prior work with such a general conversion required each user having N^2 samples making it practically meaningless and useful bounds were known only for several specific problems. This paper applies to any number m and requires $\tilde O(n/\sqrt{m})$ users. This significantly extends our understanding of the relationship of sample complexity of item level and user level DP algorithm. An additional transformation for algorithms based on the exponential mechanism is given. Along the way the paper derives a number of other interesting results such as a strong connection between so called perfect generalization and DP. The main limitation of the work is that the transformation does not lead to efficient algorithms (even if applied to an efficient algorithm).